# Algorithm-assisted interpretation of cyclic and differential pulse voltammetry for cardiac troponin detection

Wikan Danar Sunindyo[1]*, Isa Anshori[2], Kristo Abdi Wiguna[3],
Marcelus Michael Herman Kahari[3], Infall Syafalni[3,4], Latifa Dwiyanti[1,5], Uperianti[2]

1 Knowledge and Software Engineering Research Group, School of Electrical Engineering and Informatics, Institut Teknologi Bandung, Bandung, West Java, Indonesia, 2 Lab on Chip Laboratory, Biomedical Engineering Department, School of Electrical Engineering and Informatics, Institut Teknologi Bandung, Bandung, West Java, Indonesia, 3 School of Electrical Engineering and Informatics, Institut Teknologi Bandung, Bandung, West Java, Indonesia, 4 University Center of Excellence on Microelectronics, Institut Teknologi Bandung, Bandung, West Java, Indonesia, 5 Graduate School of Natural Science and Technology, Kanazawa University, Kanazawa, Japan

☻ These authors contributed equally to this work.
* wikan@informatika.org

## Abstract

Cardiovascular disease remains a leading cause of mortality worldwide, and rapid identification of cardiac biomarkers is essential for early detection. Electrochemical voltammetry techniques, particularly cyclic voltammetry (CV) and differential pulse voltammetry (DPV), are widely used for detecting cardiac troponin; however, interpretation of raw voltammetric signals is often affected by baseline drift, signal noise, and operator-dependent analysis. This study proposes an algorithm-assisted analytical framework for automated interpretation of voltammetric data obtained from a screen-printed carbon electrode potentiostat. Polynomial fitting was applied for baseline correction in CV signals, while asymmetric least squares (ALS) was employed for DPV data. Peak-to-baseline current response was extracted as a quantitative indicator of biomarker presence. The proposed method successfully identified characteristic voltammetric peaks and distinguished samples with higher and lower cardiac biomarker responses relative to a predefined detection threshold. The analysis showed close agreement with reference electrochemical analysis software, demonstrating reliable peak detection and baseline estimation. By reducing manual interpretation and improving signal clarity, the framework enhances the reproducibility and accessibility of electrochemical biosensor measurements and supports early screening of cardiac biomarkers.

## Introduction

Cardiovascular disease is a leading cause of mortality worldwide. According to the World Health Organization, cardiovascular diseases account for

**Data availability statement:** Data may be found at https://doi.org/10.6084/m9.figshare.29553506.

**Funding:** This work was supported by Institut Teknologi Bandung through the Riset PPMI STEI ITB 2024 program (No: 753/IT1.C12/TU.10/2024), awarded to Wikan Danar Sunindyo, and the Riset Unggulan ITB 2025 program (File code: DRPM.PN-6-28-2025; Contract number: 841/IT1.B07.1/TA.00/2025), awarded to Isa Anshori. The funders had no role in study design, data collection and analysis, decision to publish, or preparation of the manuscript.

**Competing interests:** The authors have declared that no competing interests exist.

approximately 18 million deaths annually, representing about one-third of global mortality [1]. Early detection relies on the identification of cardiac biomarkers such as cardiac troponin, which is released into the bloodstream during myocardial injury and is widely regarded as the gold standard biomarker for acute myocardial infarction diagnosis [2].

Electrochemical biosensors based on voltammetry have been extensively investigated as rapid and cost-effective approaches for biomarker detection. In particular, cyclic voltammetry (CV) and differential pulse voltammetry (DPV) enable sensitive electrochemical measurement of redox-active analytes and have been applied for cardiac troponin detection using screen-printed carbon electrodes [3–5].

Despite their sensitivity, interpretation of voltammetric signals remains challenging. Raw voltammograms are frequently affected by baseline drift, background current, electrode variability, and signal noise, which can obscure characteristic oxidation–reduction peaks [6]. As a result, peak identification and quantification often depend on manual inspection and operator experience, leading to variability and reduced reproducibility of electrochemical measurements [7].

Several studies have demonstrated potentiostat-based detection of cardiac troponin using electrochemical biosensors. However, limited attention has been given to automated analytical interpretation of voltammetric signals. Accurate baseline estimation and peak detection are essential because the peak-to-baseline current response serves as the primary indicator of analyte presence and concentration in electrochemical analysis [6].

To address this limitation, this study proposes an algorithm-assisted framework for automated interpretation of CV and DPV signals obtained from a screen-printed carbon electrode potentiostat. Polynomial fitting was applied for baseline correction in CV measurements, while asymmetric least squares (ALS) was used for DPV analysis. The approach aims to improve consistency of peak detection and reduce operator-dependent variability in electrochemical biosensor measurements.

## Materials and methods

### Electrochemical measurement setup

Electrochemical measurements were performed using a potentiostat connected to a screen-printed carbon electrode (SPCE) configured with a three-electrode system consisting of a carbon working electrode (WE), an Ag/AgCl reference electrode (RE), and a carbon counter electrode (CE) [4,8]. The potentiostat controlled the potential applied to the working electrode and recorded the resulting current response generated by electrochemical redox reactions.

The SPCE was selected due to its reproducibility, portability, and low cost, making it suitable for point-of-care electrochemical biosensing applications [9]. The integrated Ag/AgCl reference electrode provided a stable reference potential during measurements, while the counter electrode maintained current balance within the electrochemical cell [4,10]. Samples were introduced directly onto the electrode surface, and the current–potential response was recorded for further voltammetric analysis.

Fig 1 shows the electrochemical measurement hardware used in this study.

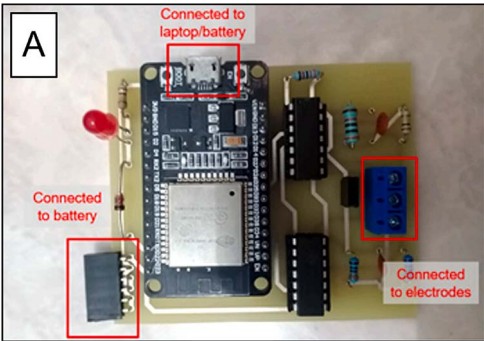

**Fig 1. Example of potentiostat hardware [11].**

## Voltammetric measurement

Voltammetry measurements were performed to analyze electrochemical responses of the analyte by recording the current as a function of the applied potential. During measurement, a controlled potential was applied to the working electrode relative to the reference electrode, and the resulting current generated by oxidation–reduction reactions at the electrode surface was recorded. The measured current corresponds to Faradaic processes associated with electroactive species, while background current arises from capacitive effects at the electrode interface [4,7].

The resulting current–potential relationship (voltammogram) was used to identify characteristic peak responses associated with the presence of the target analyte. These peaks served as the basis for subsequent signal processing and quantitative interpretation.

**Cyclic voltammetry (CV) measurement.** Cyclic voltammetry (CV) was performed to evaluate the electrochemical behavior of the analyte by monitoring oxidation–reduction processes at the electrode surface. During the measurement, a linearly varying potential was applied to the working electrode relative to the reference electrode, and the resulting current response was recorded continuously. The potential was swept between predefined limits in forward and reverse directions to generate a current–potential curve (voltammogram).

The resulting anodic and cathodic peak currents were used to identify electroactive reactions occurring at the electrode interface. These peak characteristics provided qualitative information on the presence of the analyte and served as a reference for subsequent signal processing and comparison with differential pulse voltammetry measurements [4,7].

**Differential pulse voltammetry (DPV) measurement.** Differential pulse voltammetry (DPV) was employed for biomarker detection due to its high sensitivity and improved signal-to-noise characteristics for low-concentration analytes. In this technique, a sequence of potential pulses was superimposed on a slowly increasing base potential applied to the working electrode relative to the reference electrode. The current was sampled immediately before each pulse and at the end of the pulse, and the difference between these two measurements was recorded as the DPV response.

This differential measurement suppresses capacitive background current and enhances Faradaic current originating from oxidation–reduction reactions at the electrode surface. The resulting voltammogram exhibited well-defined peak currents corresponding to electroactive species. The peak current relative to the baseline was used as the analytical signal for identifying the presence of the target analyte and for subsequent signal-processing analysis [12–14].

## Sensor reader and data acquisition

The sensor reader used in this study was adapted from the low-cost potentiostat design reported by Anshori et al. [1,11,12]. The device was connected to a screen-printed carbon electrode (SPCE) and used to acquire electrochemical signals from blood samples containing cardiac biomarkers. During measurement, the potentiostat applied a controlled

potential to the working electrode and recorded the resulting current generated by oxidation–reduction reactions at the electrode surface.

The current response was measured over a defined potential range and stored as a current–potential dataset (voltammogram). This voltammetric signal served as the raw data for analysis. The recorded signals contained both background current and Faradaic current originating from electroactive species. Therefore, further processing was required to identify characteristic peak responses and baseline levels associated with the presence of the target analyte.

All recorded measurements were exported as numerical data files for subsequent signal processing and peak analysis. Fig 2 illustrates the potentiostat hardware used for electrochemical data acquisition.

**Input data representation.** Electrochemical measurements produced current–potential datasets obtained from cyclic voltammetry (CV) and differential pulse voltammetry (DPV) experiments. Each dataset consisted of paired numerical values representing the applied potential (V) and the measured current (μA), forming a voltammogram.

For CV measurements, multiple scans were recorded, and the scan obtained under stable electrode conditions was selected for analysis. Oxidation and reduction peaks were identified from the current–potential curve, and their peak currents were evaluated relative to the estimated baseline.

For DPV measurements, a single current–potential dataset was recorded for each sample. Analytical information was derived from the peak current relative to the baseline current, which serves as the primary electrochemical indicator of analyte presence and concentration. The peak-to-baseline difference (current response) was calculated and used for comparison between measurement samples.

Because voltammetric signals include background (capacitive) current and Faradaic current generated by electroactive species, baseline estimation was required before peak extraction. Therefore, signal processing procedures were applied to identify characteristic peak responses and to distinguish electrochemical signals associated with the target analyte from background noise [6,12].

**Signal processing and data analysis.** Baseline correction and peak extraction were performed to interpret voltammetric signals obtained from cyclic voltammetry (CV) and differential pulse voltammetry (DPV) measurements. Raw voltammograms contain both Faradaic current, originating from oxidation–reduction reactions of electroactive species, and non-Faradaic (capacitive) background current. Therefore, baseline estimation was required prior to quantitative analysis [6,12].

For CV data, baseline drift was estimated using polynomial fitting applied to regions without significant electrochemical activity. The fitted baseline was subtracted from the original signal to obtain a corrected current profile. Oxidation and reduction peaks were then identified from the corrected voltammogram, and the peak current relative to the baseline was extracted as the analytical parameter [7].

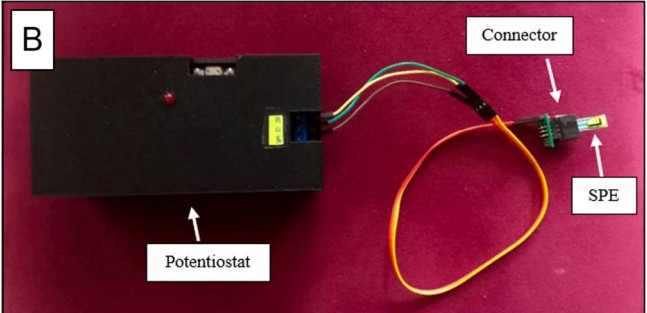

**Fig 2. Potensiostat hardware used for the sensor reader [11].**

For DPV data, asymmetric least squares (ALS) baseline correction was applied to remove background current and improve signal clarity [15]. After baseline subtraction, peak detection was performed to identify the potential at which the maximum Faradaic response occurred. The peak-to-baseline current difference (peak current) was used as the electrochemical response value and compared across samples.

In both measurement modes, the peak current relative to the baseline was used as an indicator of analyte presence because it reflects the electrochemical reaction occurring at the electrode interface. DPV measurements were emphasized for biomarker detection due to their higher sensitivity and improved signal-to-noise ratio for low-concentration analytes [16]. The analysis relied on deterministic signal-processing procedures rather than inferential statistical modeling, focusing on reproducible extraction of electrochemical peak characteristics from the recorded voltammograms.

## Computational processing of voltammetric data

The electrochemical signals obtained from voltammetry measurements were digitally processed prior to analysis. The recorded current–potential datasets were imported into a computational analysis environment for signal conditioning and feature extraction. Processing was performed to remove background current, estimate the signal baseline, and identify characteristic peak responses associated with electrochemical reactions of the analyte.

The computational procedures were designed to ensure consistent interpretation of voltammograms by applying algorithmic baseline correction and peak detection rather than manual inspection. The extracted peak current relative to the baseline was used as the primary analytical parameter for evaluating electrochemical responses [6,12].

## Data processing workflow

The analytical workflow used in this study is illustrated in Fig 3. Electrochemical measurements were first obtained using a potentiostat connected to the sensor electrode. The instrument generated current–potential datasets, which were exported as numerical files containing applied potential (V) and measured current (µA) [12].

The recorded datasets were then subjected to computational processing. The voltammetric signals underwent baseline correction to remove background (capacitive) current [6], followed by peak detection to identify characteristic electrochemical responses. The extracted peak current relative to the baseline was used as the analytical parameter for evaluating the presence of the target analyte.

This workflow enabled consistent processing of voltammograms from acquisition to quantitative signal interpretation. Fig 3 presents the processing sequence from electrochemical measurement to peak extraction and analysis.

**Analytical processing functions.** Based on the analytical workflow shown in Fig 3, a set of processing functions was defined for handling voltammetric data. These functions describe the sequence of computational operations required to transform raw current–potential measurements into interpretable electrochemical signals. The processing steps include data import, baseline correction, peak detection [6], and extraction of peak current values from the voltammogram [12]. The implemented functions correspond to standard procedures used in electrochemical signal analysis and were applied consistently to all recorded measurements. The detailed processing operations are summarized in Table 1.

## Computational processing architecture

The recorded voltammetric datasets were processed using a modular computational workflow designed for electrochemical signal analysis. The processing architecture consisted of data acquisition, preprocessing, baseline correction, and peak extraction modules. Each module operated sequentially to transform raw current–potential measurements into analytical parameters.

The preprocessing stage handled numerical input data obtained from the potentiostat and organized the datasets into structured current–potential arrays. Baseline estimation was then performed to remove background (capacitive) current

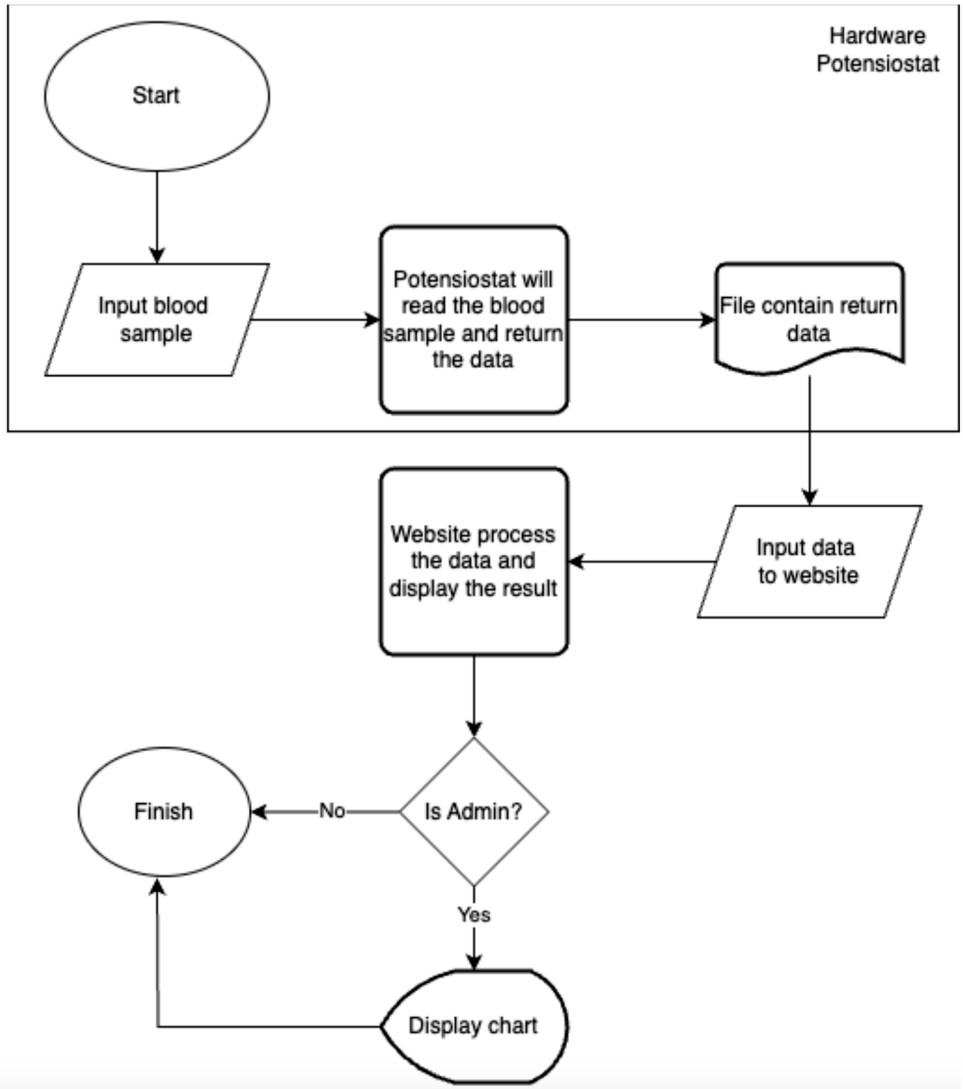

**Fig 3. Flowchart diagram of software system.**

[6], followed by peak detection to identify characteristic Faradaic responses occurring at the electrode interface [12]. The extracted peak current relative to the baseline was used as the primary electrochemical response variable [12].

For differential pulse voltammetry data, baseline correction was implemented using asymmetric least squares (ALS) fitting [15]. This modular structure ensured reproducible processing of voltammograms and enabled consistent interpretation of electrochemical signals across all measurements. Fig 4 illustrates the data-processing architecture applied in this study.

## Computational environment

The voltammetric datasets were processed within a computational analysis environment designed to handle numerical electrochemical data. The processing system received measurement datasets generated by the potentiostat and performed automated preprocessing, baseline correction, and peak extraction.

**Table 1. Functional requirements of software system.**

| ID | Requirement | Description |
|---|---|---|
| F01 | The system should accept user information from users who want to create an account. | Users can create an account by providing an email and password. |
| F02 | The system should accept login information from users who want to sign in. | Users can log in using their email and password. |
| F03 | The system should accept potentiostat data from users through the website. | Users can upload potentiostat data to the website. |
| F04 | The system should display predictions of heart disease potential for users. | Users can view the predicted potential for heart disease based on data obtained from a potentiostat device. |
| F05 | The system should allow changing a regular account to a lab technician account and vice versa. | Admins can convert a regular account to a lab technician account and vice versa on the designated page. |
| F06 | The system should display the test history for each user. | Regular users can view the history and results of tests they have conducted. |
| F07 | For lab technicians and admins, the system should display the test history of all users. | Lab technicians and admins have access to the test history of all users. |
| F08 | The system should display graphical visualizations of data captured by the potentiostat device. | Lab technicians and admins can view and analyze graphical visualizations of data captured by the device. |

The computational workflow enabled consistent handling of measurement data and minimized manual intervention during signal interpretation. Numerical data were stored and organized for subsequent analysis and comparison between measurements. The analysis procedures focused on extraction of electrochemical features, particularly peak current relative to the baseline, which served as the analytical response parameter [12].

The computational environment was implemented to ensure reliable processing of multiple datasets and reproducible interpretation of voltammograms across repeated measurements.

**Visualization of electrochemical signals.** To facilitate interpretation of electrochemical measurements, the processed voltammetric data were visualized as current–potential plots (voltammograms). The visualization allowed observation of baseline behavior and identification of oxidation–reduction peaks obtained after signal processing. Graphical representation of voltammograms is commonly used in electrochemical analysis to evaluate peak current and electrochemical response characteristics [7].

The visualization served only as a representation of the processed analytical data and did not affect the electrochemical measurement or signal-processing procedures.

**Visualization of voltammetric signals.** Processed electrochemical data were represented as current–potential plots (voltammograms) to facilitate interpretation of electrochemical responses. The voltammogram displays the measured current as a function of applied potential, allowing identification of oxidation and reduction peaks corresponding to electroactive species at the electrode surface. Visualization of the corrected signal enabled verification of baseline estimation and peak extraction results.

The graphical representation was used solely to illustrate analytical outcomes and did not affect the electrochemical measurement or signal-processing procedures. Visualization of voltammograms is a standard practice in electrochemical analysis for evaluating peak current and electrochemical behavior of analytes [7].

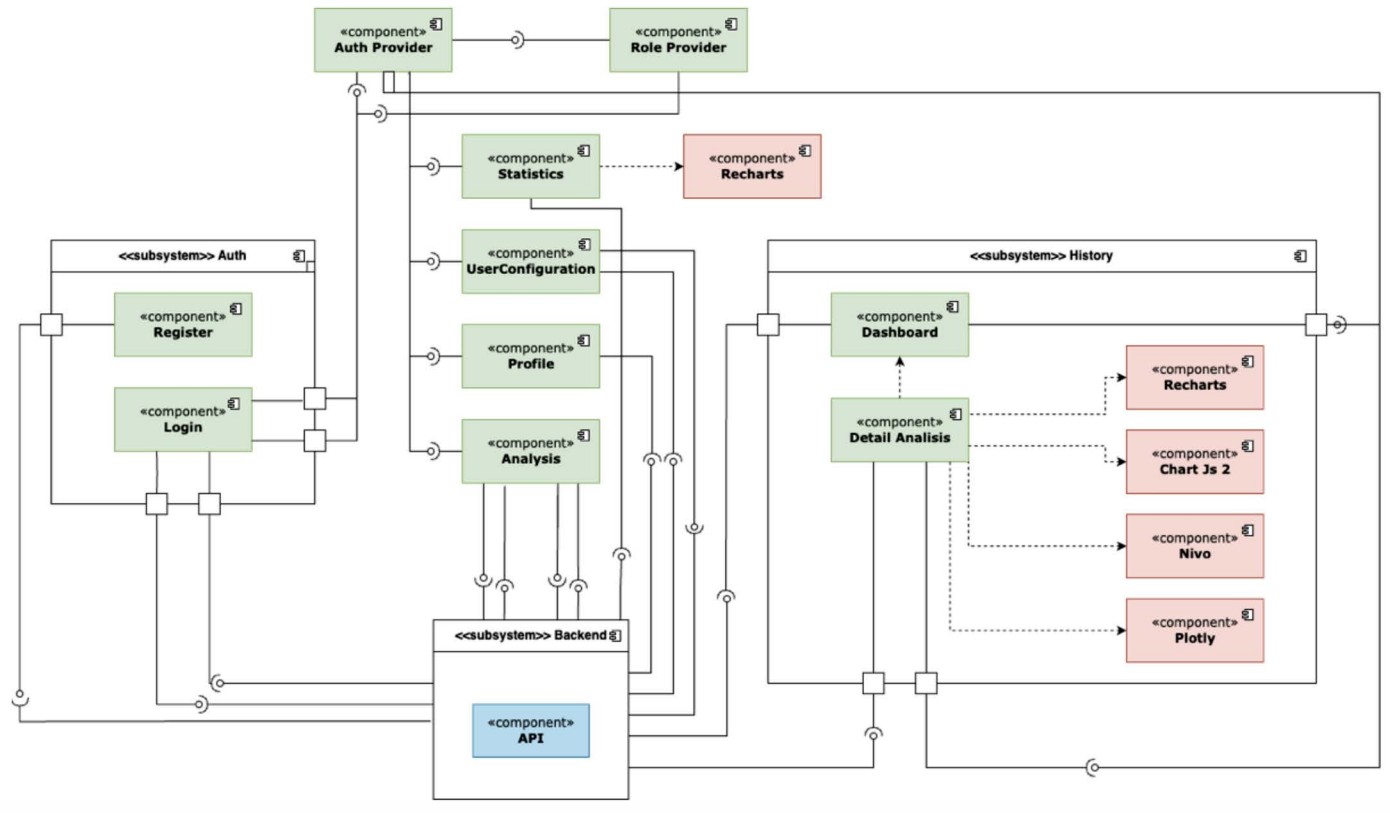

**Fig 4. Component diagram for the front end system.**

**Identification of electrochemical features.** To support interpretation of the processed voltammograms, key electrochemical features were indicated on the current–potential plots. The baseline level and detected peak positions were marked to show the location of characteristic oxidation–reduction responses obtained after signal processing. These markers correspond to analytical parameters commonly used in voltammetric analysis, including peak potential and peak current.

Indicating these electrochemical features enabled verification of peak detection and baseline estimation results without altering the analytical calculations. Marking peak current and peak potential is a standard approach in voltammetry for interpreting electrochemical reactions and evaluating analyte response [7].

**Visual Inspection of Processed Signals.** Processed voltammetric data were examined through graphical inspection of the current–potential plots. The visualization allowed detailed observation of baseline behavior and peak characteristics after signal processing. This inspection was used to confirm that detected peaks corresponded to electrochemical reactions rather than background fluctuations.

The graphical inspection served as a verification step for the computational analysis by allowing examination of peak position and peak height relative to the baseline. Visual examination of voltammograms is commonly employed in electrochemical studies to validate automated peak detection and to ensure correct interpretation of electrochemical responses [7].

**Graphical representation of voltammetric data.** Processed voltammetric signals were displayed as current–potential plots to illustrate electrochemical responses obtained after signal processing. The graphical representation enabled

visualization of baseline behavior and identification of oxidation and reduction peaks corresponding to electroactive species.

The plots were generated using computational visualization tools to represent the analytical results without modifying the underlying measurement data. The graphical output served only as a representation of the processed electrochemical signals and did not influence the electrochemical measurement or signal-processing procedures. Graphical presentation of voltammograms is a standard approach in electrochemical analysis for interpreting peak current and electrochemical behavior of analytes [7].

## Results and discussion

Electrochemical measurements obtained from cyclic voltammetry (CV) and differential pulse voltammetry (DPV) produced current–potential curves (voltammograms) that contained background drift and signal noise. After applying baseline correction and peak extraction procedures, characteristic electrochemical peaks became distinguishable from the background signal. The corrected voltammograms enabled identification of peak currents associated with electroactive species present in the samples.

The extracted peak current relative to the baseline served as the analytical response parameter. Samples exhibiting higher peak current responses showed stronger electrochemical activity compared with samples with lower responses. DPV measurements provided clearer peak separation and improved signal-to-noise ratio compared with CV, making them more suitable for detecting low-concentration electroactive analytes.

These findings demonstrate that algorithm-assisted processing improves interpretation of voltammetric signals by reducing ambiguity caused by baseline drift and noise. The approach enables consistent extraction of electrochemical features from raw measurement data and supports screening of cardiac biomarker responses.

### Voltammetric data analysis

Raw voltammetry measurements produced current–potential curves containing background drift and measurement noise. Therefore, baseline correction and peak identification were performed prior to quantitative evaluation. Several baseline correction approaches, including polynomial fitting and asymmetric least squares (ALS), were evaluated to obtain a stable estimation of the background signal. After baseline subtraction, peak detection was conducted to locate the maximum Faradaic current corresponding to electroactive species. The peak-to-baseline current difference was extracted as the electrochemical response parameter.

Differential pulse voltammetry (DPV) measurements were used for analysis due to their higher sensitivity and clearer peak separation compared with cyclic voltammetry. For each dataset, the peak potential and peak current were determined from the corrected voltammogram. The analytical response was defined as the peak current relative to the baseline current.

Two representative DPV measurements were compared to evaluate differences in electrochemical response. The response values obtained from the measurements were contrasted to determine relative electrochemical activity between samples. The processed voltammograms and extracted peak characteristics are presented in Figs 5 and 6.

For each differential pulse voltammetry (DPV) measurement, the analytical parameter was defined as the peak current relative to the estimated baseline. The peak location was characterized by its potential coordinate (v) and current value (c), while the baseline level (b) was determined from the corrected voltammogram. The electrochemical response value (d) was calculated as the vertical distance between the peak current and the baseline.

To compare measurements, the response values obtained from different samples were evaluated relative to a predefined response threshold. This threshold was used to categorize the electrochemical responses into lower-response and higher-response groups. Samples with response values below the threshold exhibited weak electrochemical activity, whereas samples with response values above the threshold showed stronger electrochemical activity.

                                                    

```
{
  "voltage": [-0.907936, -0.897728, -0.887521, -0.877313, -0.867106, -0.85
-0.744615, -0.734407, -0.7242, -0.713992, -0.703785, -0.693577, -0.683369,
-0.571086, -0.560879, -0.550671, -0.540463, -0.530256, -0.520048, -0.50984
-0.397557, -0.38735, -0.377142, -0.366935, -0.356727, -0.34652, -0.336312,
-0.224029, -0.213821, -0.203614, -0.193406, -0.183199, -0.172991, -0.16278
-0.0607082, -0.0505006, -0.040293, -0.0300855, -0.0198779, -0.00967033, 0.
0.102612, 0.11282, 0.123028, 0.133235, 0.143443, 0.15365, 0.163858, 0.1740
0.296556, 0.306764, 0.316971, 0.327179, 0.337387, 0.347594, 0.357802, 0.36
0.4905, 0.500708, 0.510915, 0.521123, 0.53133, 0.541538, 0.551746, 0.56195
0.684444, 0.694652, 0.704859, 0.715067, 0.725274, 0.735482, 0.745689, 0.75
0.878388, 0.888595, 0.898803, 0.90901, 0.919218, 0.929426, 0.939633, 0.949
  "current": [121.295, 120.679, 119.615, 119.783, 119.503, 118.719, 118.55
117.137, 117.151, 117.095, 117.095, 116.983, 117.151, 116.871, 116.815, 11
116.983, 117.095, 116.647, 116.759, 117.039, 116.843, 116.962, 117.011, 11
117.935, 117.823, 118.019, 118.719, 118.943, 118.831, 119.755, 120.203, 12
124.459, 124.795, 125.593, 126.027, 126.083, 127.511, 131.347, 137.647, 14
137.871, 135.855, 133.867, 131.571, 129.331, 127.007, 123.759, 120.259, 11
105.419, 105.475, 104.831, 104.299, 104.187, 103.459, 102.367, 102.479, 10
99.5676, 99.2596, 98.7836, 99.2876, 99.2036, 99.1826, 99.0076, 98.9253, 98
100.099, 100.358, 100.491, 100.771, 100.575, 100.995, 101.611, 101.611, 10
108.555],
  "baseline": [
    [-0.907936, 119.381641846171],
    [-0.897728, 119.25694733595],
    [-0.887521, 119.132252844862],
    [-0.877313, 119.007558406262],
    [-0.867106, 118.882864058332],
    [-0.856898, 118.758169847007],
    [-0.846691, 118.633475824424],
    [-0.836483, 118.508782003944],
    [-0.826275, 118.384088317274],
    [-0.816068, 118.259394697105],
    [-0.80586, 118.13470085329],
    [-0.795653, 118.010005897331],
    [-0.785445, 117.885308798466],
```

**Fig 5. Result of analysis on the measured data (1).**

```
],
"info": {
  "v": 0.0617827,
  "c": 153.215,
  "b": [0.0617827, 107.122255297897],
  "d": 46.0927447021026
}
```

**Fig 6. Result of analysis on the measured data (2).**

The classification reflects differences in electrochemical behavior associated with the presence of cardiac biomarker species rather than a clinical diagnosis. Table 2 summarizes the relationship between the response value and the corresponding electrochemical response category derived from the analysis.

Fig 7 presents two representative differential pulse voltammetry (DPV) curves obtained from different samples. The first sample exhibits a smaller peak current relative to the baseline, indicating a weaker electrochemical response. In contrast, the second sample shows a higher peak current above the baseline, reflecting stronger electrochemical activity.

The separation between the two response levels demonstrates that the applied signal-processing procedure enables clear identification of peak features in the voltammograms. The higher response is consistent with increased electrochemical activity associated with the presence of cardiac biomarker species, whereas the lower response corresponds to weaker electrochemical activity.

**Table 2. Comparison for threshold and current response.**

| Comparison | Disease Likelihood | Result |
|---|---|---|
| Current response > threshold | High potential | True |
| Current response <= threshold | Low potential | False |

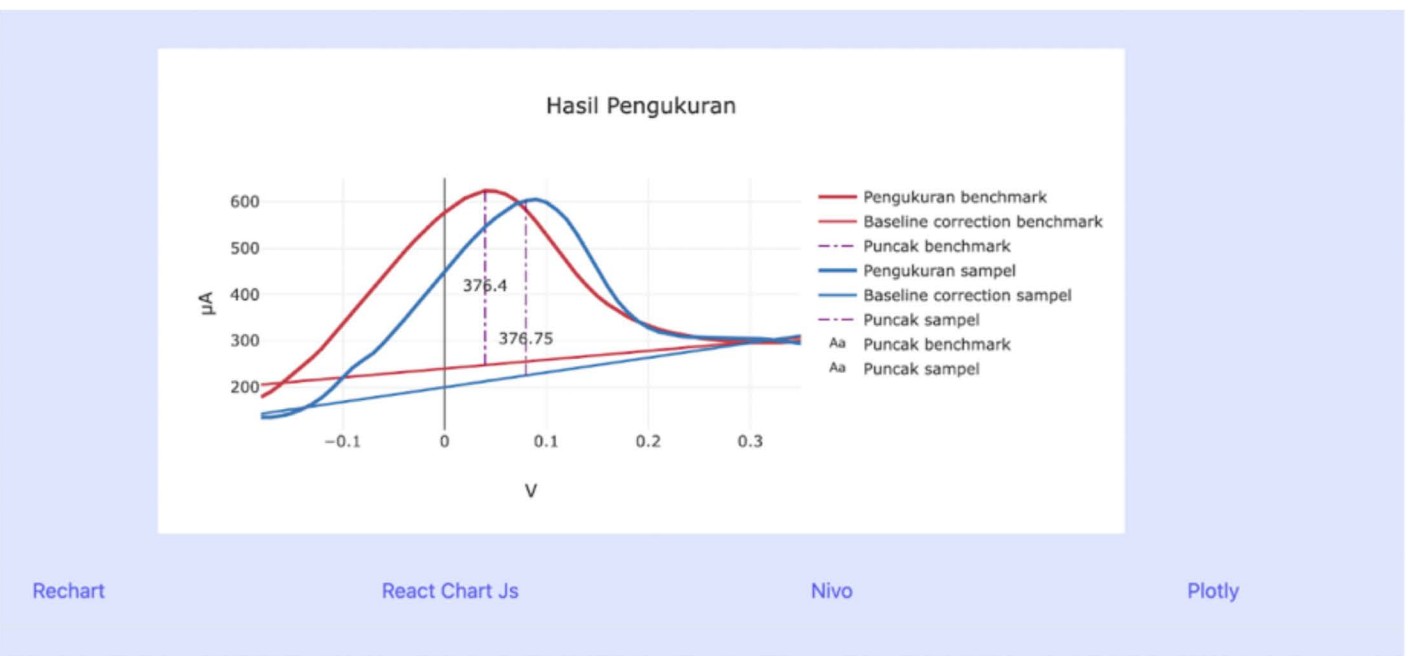

**Fig 7. Analysis of the voltammetry data result.**

These results indicate that DPV measurements combined with baseline correction and peak extraction allow reliable differentiation of electrochemical response levels in the analyzed samples.

## Data storage structure

Each voltammetry measurement consisted of raw electrochemical data and the corresponding processed analytical results. The stored information included the measurement configuration (cyclic voltammetry or differential pulse voltammetry), baseline estimation, peak location, and peak-to-baseline response value. To ensure traceability of the analysis, the measurement data and processed parameters were organized in a structured data model, illustrated in Fig 8.

A relational data storage structure was used to maintain consistent linkage between raw measurements and extracted analytical features. This organization allowed each processed result to be traced back to its original measurement and enabled systematic comparison of electrochemical responses relative to a predefined response threshold.

Raw voltammetry measurements were stored as comma-separated datasets containing potential and current values for each measurement. For cyclic voltammetry (CV), a single dataset was sufficient for analysis because the complete current–potential response was contained in one measurement sequence. In contrast, differential pulse voltammetry (DPV) required two datasets to enable comparison between reference and sample responses.

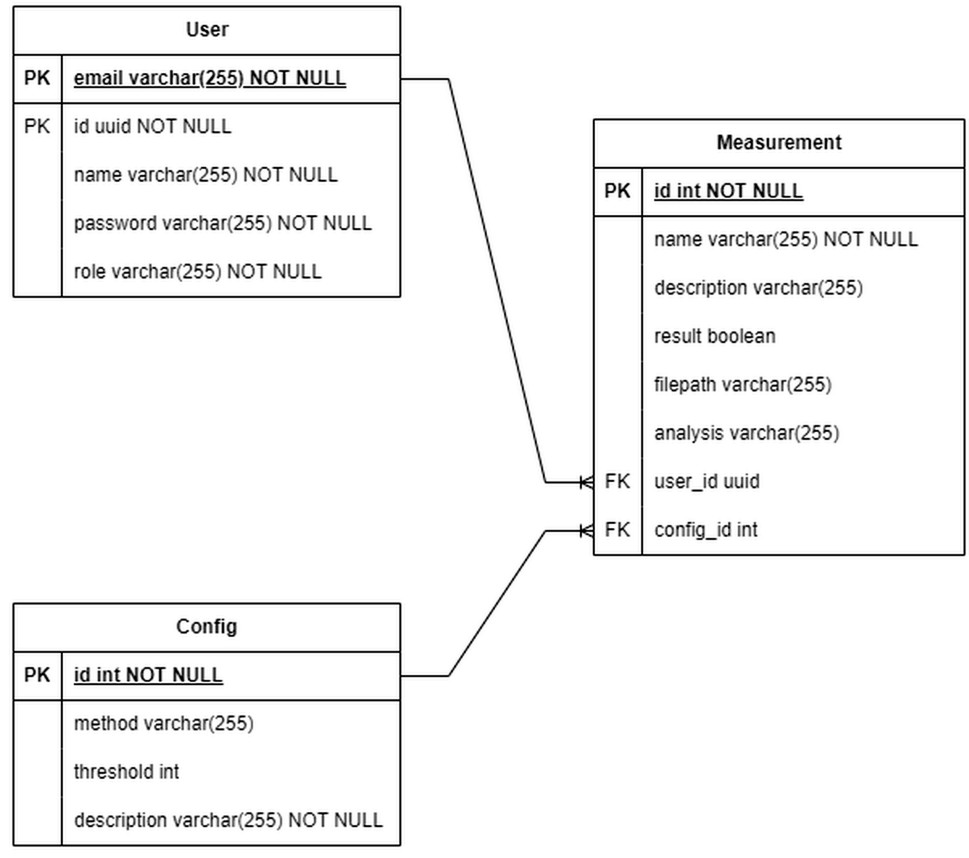

**Fig 8. Data organization schema.**

Each dataset was processed to obtain baseline estimation, peak location, and peak-to-baseline response values. The processed analytical parameters were then linked to the corresponding measurement data to ensure traceability of the analysis. The response values were subsequently evaluated relative to a predefined response threshold in order to categorize the electrochemical activity level of the sample. This procedure allowed consistent comparison between measurements without implying clinical diagnosis, and provided a structured interpretation of the electrochemical signals.

## Voltammetric signal processing algorithms

Voltammetry measurements produce current–potential curves that often contain background drift and measurement noise. Therefore, signal processing was required before quantitative interpretation. Baseline correction algorithms were applied to estimate and remove the non-Faradaic background current. In this study, polynomial fitting and asymmetric least squares (ALS) methods were used to obtain a stable baseline estimation.

After baseline subtraction, peak detection was performed to identify the Faradaic current corresponding to electroactive species. The peak height relative to the corrected baseline was extracted as the analytical response parameter. To improve signal clarity, smoothing procedures such as Savitzky–Golay filtering were applied prior to peak detection to reduce high-frequency noise while preserving peak shape.

These processing steps enabled reliable determination of peak potential and peak current from the voltammograms, which are standard parameters used to interpret electrochemical measurements.

**Cyclic voltammetry signal processing.** Cyclic voltammetry (CV) measurements were analyzed to identify oxidation and reduction peaks from the recorded current–potential curves. Prior to peak detection, baseline correction was performed to remove background current associated with capacitive and non-Faradaic processes. A second-order polynomial fitting method was applied to estimate the baseline across the potential range, and the estimated baseline was subtracted from the measured current to obtain a corrected signal.

Following baseline correction, peak detection was conducted to identify local maxima and minima corresponding to oxidation and reduction reactions. For each detected peak, the peak potential and peak current were extracted, and the peak height relative to the baseline was calculated as the analytical parameter representing the electrochemical response.

An example implementation of the analysis procedure is provided in Fig 9 to illustrate the computational workflow used in this study. The code serves as a reference implementation and does not replace the mathematical description of the method.

The cyclic voltammetry dataset consisted of paired potential and current values obtained from the measurement. Baseline estimation was first performed using a second-order polynomial fitting applied to the current–potential data. The estimated baseline was subsequently subtracted from the measured current to obtain a baseline-corrected signal suitable for peak analysis.

To improve baseline accuracy in different electrochemical regions, linear fitting was additionally applied to the initial and final portions of the voltammogram corresponding to oxidation and reduction segments. After baseline correction, peak detection was carried out on the corrected signal to identify local maxima and minima representing oxidation and reduction reactions.

For each detected peak, the peak potential and peak current were determined, and the peak height relative to the baseline was calculated. These extracted parameters describe the electrochemical response and were used for subsequent analysis of the voltammetric measurements.

**Differential pulse voltammetry signal processing.** Differential pulse voltammetry (DPV) measurements were analyzed to extract electrochemical peak responses from the recorded current–potential data. Because DPV signals commonly contain background drift and asymmetric noise, baseline correction was performed prior to peak evaluation. An asymmetric least squares (ALS) method was applied to estimate the baseline by iteratively fitting a smooth curve to the non-Faradaic background current. The estimated baseline was then subtracted from the measured signal to obtain a corrected voltammogram.

Following baseline correction, peak detection was carried out to identify characteristic Faradaic peaks corresponding to electroactive species. For each peak, the peak potential and peak current were determined, and the peak height relative to the baseline was calculated as the analytical response parameter.

The extracted peak parameters were used to evaluate differences in electrochemical response between samples. An example implementation of the DPV signal-processing procedure is illustrated in Fig 10.

The differential pulse voltammetry data consisted of paired potential and current values obtained from the measurement. Prior to analysis, the initial portion of the signal was excluded to minimize transient effects and measurement noise commonly present at the beginning of the acquisition. Baseline estimation was then performed using an asymmetric least squares (ALS) approach to model the non-Faradaic background current. The estimated baseline was subtracted from the measured current to obtain a baseline-corrected voltammogram.

Peak detection was subsequently applied to the corrected signal to identify the dominant Faradaic peak associated with electroactive species. For the detected peak, the peak potential and peak current were determined, and the peak height relative to the baseline was calculated as the analytical response parameter. These extracted values were used to characterize the electrochemical response in the DPV measurements.

## Computational processing environment

The signal-processing procedures were implemented in a centralized computational environment to ensure consistent analysis of voltammetry measurements. The processing workflow separated data acquisition from data analysis, allowing

```
1   def analyze_cv(split_columns):
2       voltage = split_columns[f'V2']
3       current = split_columns[f'µA2']
4
5       try:
6           polynomial_coefficients = np.polyfit(voltage, current, deg=2)
7           polynomial = np.poly1d(polynomial_coefficients)
8           baseline_current = polynomial(voltage)
9           baseline_subtracted_current = current - baseline_current
10      except Exception as e:
11          print(f"Error: {e}")
12
13      baseline_oxidation_start = 0
14      baseline_oxidation_end = int(len(voltage) * 0.9)
15      baseline_oxidation_fit = np.polyfit(
16          voltage[baseline_oxidation_start:baseline_oxidation_end],
17          current[baseline_oxidation_start:baseline_oxidation_end], 1)
18      baseline_oxidation = np.polyval(baseline_oxidation_fit, voltage)
19      res_baseline_oxidation = np.column_stack((voltage, baseline_oxidation))
20
21      baseline_reduction_start = -int(len(voltage) * 0.1)
22      baseline_reduction_end = -1
23      baseline_reduction_fit = np.polyfit(
24          voltage[baseline_reduction_start:baseline_reduction_end],
25          current[baseline_reduction_start:baseline_reduction_end], 1)
26      baseline_reduction = np.polyval(baseline_reduction_fit, voltage)
27      res_baseline_reduction = np.column_stack((voltage, baseline_reduction))
28
29      oxidation_peaks, oxidation_properties = find_peaks(baseline_subtracted_current, height=0)
30      reduction_peaks, reduction_properties = find_peaks(-baseline_subtracted_current, height=0)
31
32      if oxidation_peaks.size > 0:
33          highest_oxidation_peak = oxidation_peaks[np.argmax(oxidation_properties["peak_heights"])]
34          highest_oxidation_peak_voltage = voltage.iloc[highest_oxidation_peak]
35          highest_oxidation_peak_current = current[highest_oxidation_peak]
36          oxidation_baseline_current_at_peak = baseline_oxidation[highest_oxidation_peak]
37          oxidation_peak_to_baseline_current_difference = highest_oxidation_peak_current - oxidation_baseline_current_at_peak
38          oxidation_info = {
39              "v": highest_oxidation_peak_voltage,
40              "c": highest_oxidation_peak_current,
41              "b": oxidation_baseline_current_at_peak,
42              "d": oxidation_peak_to_baseline_current_difference
43          }
44      if reduction_peaks.size > 0:
45          highest_reduction_peak = reduction_peaks[np.argmax(reduction_properties["peak_heights"])]
46          highest_reduction_peak_voltage = voltage.iloc[highest_reduction_peak]
47          highest_reduction_peak_current = current[highest_reduction_peak]
48          reduction_baseline_current_at_peak = baseline_reduction[highest_reduction_peak]
49          reduction_peak_to_baseline_current_difference = highest_reduction_peak_current - reduction_baseline_current_at_peak
50          reduction_info = {
51              "v": highest_reduction_peak_voltage,
52              "c": highest_reduction_peak_current,
53              "b": reduction_baseline_current_at_peak,
54              "d": reduction_peak_to_baseline_current_difference
55          }
56      return voltage, current, res_baseline_oxidation, res_baseline_reduction, oxidation_info, reduction_info
```

**Fig 9. Example implementation of the CV signal-processing algorithm.**

```python
def analyze_dpv(split_columns) :
    voltage = split_columns['V']
    current = split_columns['µA']

    voltage = trim_head(voltage, trim_ratio=0.05)
    current = trim_head(current, trim_ratio=0.05)
    baseline_current = baseline_als(current, 1E6, 0.01)
    baseline_subtracted_current = current - baseline_current
    baseline = np.column_stack((voltage, baseline_current))

    peaks, properties = find_peaks(baseline_subtracted_current, height=0)
    if peaks.size > 0:
        highest_peak = peaks[np.argmax(properties["peak_heights"])]
        highest_peak_voltage = voltage.iloc[highest_peak]
        highest_peak_current = current.iloc[highest_peak]
        baseline_current_at_peak = baseline[highest_peak]
        peak_to_baseline_current_difference = highest_peak_current - baseline_current_at_peak[1]
        info = {
            "v": highest_peak_voltage,
            "c": highest_peak_current,
            "b": baseline_current_at_peak.tolist(),
            "d": peak_to_baseline_current_difference
        }
    else:
        info = {"v": None, "c": None, "b": None, "d": None}
    return voltage, current, baseline, info
```

**Fig 10. Example implementation of the DPV signal-processing algorithm.**

raw measurement data to be submitted to a processing unit where baseline correction, peak detection, and response extraction were performed.

This architecture enabled computationally intensive operations to be executed independently of the measurement device, ensuring stable processing performance and reproducible analytical results across different measurement sessions. The overall processing workflow is illustrated in Fig 11.

The computational processing environment consisted of an analysis module and a structured data repository. Raw voltammetry measurements were submitted to the processing module, where baseline correction, peak detection, and response extraction were performed. The processed results were linked to the corresponding raw measurement data to maintain traceability of the analysis.

A relational database was used to store measurement metadata and extracted analytical parameters, while the raw and processed datasets were archived in external storage to preserve the original measurement records. This organization enabled repeated access to previously processed measurements without reprocessing the raw data and ensured consistent evaluation across different measurement sessions.

## Data submission and processing workflow

Measurement data were provided to the analysis system in the form of comma-separated files containing potential and current values. For cyclic voltammetry (CV), a single dataset was sufficient for analysis, whereas

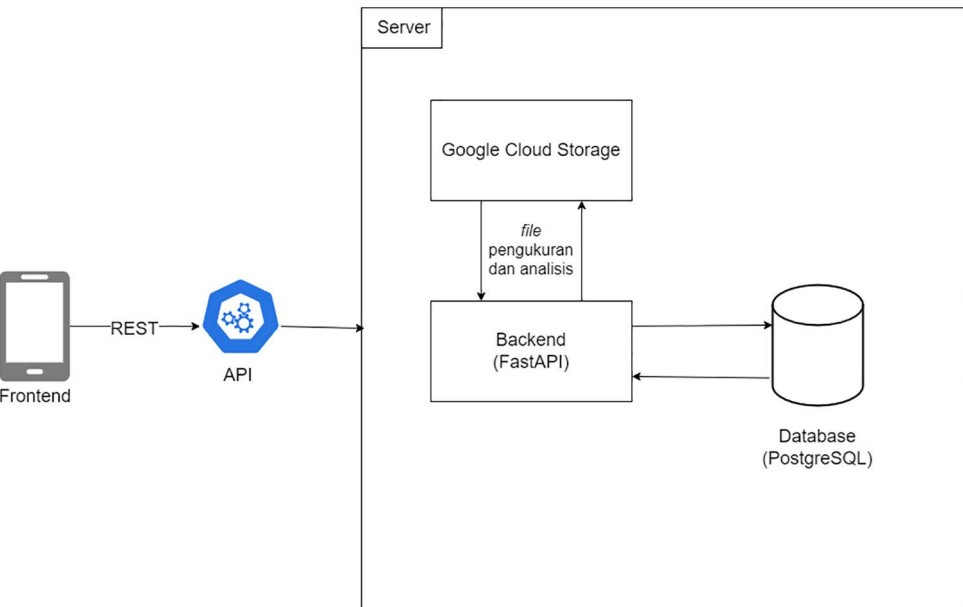

**Fig 11. Processing workflow for voltammetric data analysis.**

differential pulse voltammetry (DPV) required two datasets to enable comparison between reference and sample responses.

After submission, the datasets were validated to ensure appropriate format and completeness before analysis. The validated data were then processed through the analysis pipeline, which included baseline correction, peak detection, and extraction of peak parameters. Upon completion of processing, the analytical results were returned and presented as processed voltammograms with the corresponding extracted peak characteristics, as illustrated in Fig 12.

### Result archiving and retrieval

Processed measurements were archived together with their corresponding analytical parameters to enable traceability of the analysis. Each stored record included the original voltammetry dataset and the extracted peak characteristics obtained after signal processing. The archived results could be retrieved to review the processed voltammograms and verify previously analyzed measurements without repeating the computational procedure.

The stored records provided a consistent history of analyzed measurements and supported comparison between different datasets. An example of retrieved analysis output is shown in Fig 13.

**Chart page.** For the chart system, we compared between four open-source libraries to see which library is most suitable for this case. The four libraries were: Recharts, Chart Js, Nivo, and Plotly. Because there are two types of data that are displayed (CV and DPV), the result is presented in a table to make it easier to read.

**Chart testing.** Chart testing was conducted using one data file each for CV and DPV data. The data file used to test CV was 'CV Ascorbid Acid - 2.csv', while the data file used for testing DPV was 'DPV Ascorbic Acid - 1.csv'. The testing involved displaying the charts for the CV and the DPV data on the website using the four previously defined frameworks, respectively. The testing environment was a blank webpage that only contained the chart displays to minimize distractions from external influences. Additionally, testing was conducted by comparing the output charts with benchmark charts obtained from the PSTrace software. This comparison was done to determine whether the output from the proposed software is accurate and how it performs relative to to the benchmark software.

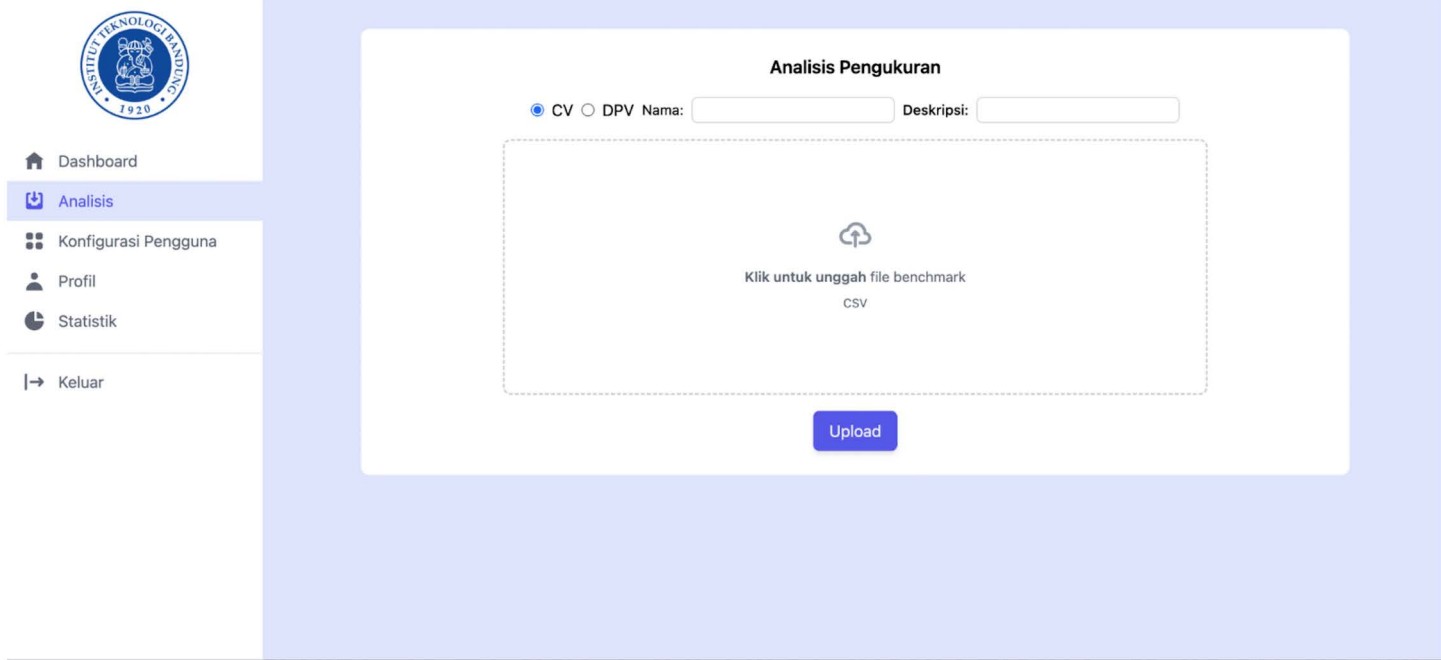

**Fig 12. Example output of processed voltammetry analysis.**

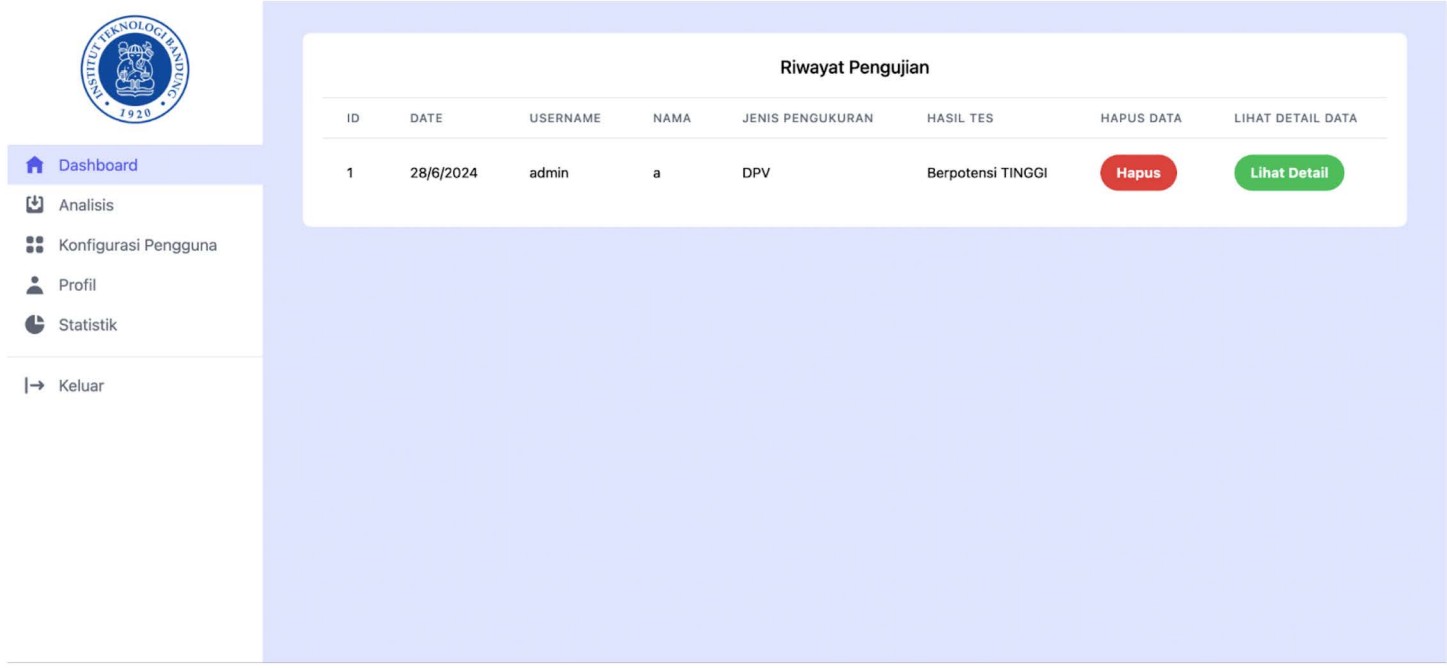

**Fig 13. Example of archived voltammetry analysis output.**

The testing used the factors of chart completeness based on the material in the Chart section. These factors are:

1. Annotation

2. Interactivity

3. Parameter manipulation

The annotation factor indicates whether the chart allows for the addition of reference lines. The purpose of reference lines is to assist the user in seeing the difference between the peak and baseline values. This value is useful for determining whether an individual is at high or low risk of heart disease.

The interactivity factor indicates whether a text box containing an explanation of the actual values appears when a user hovers over the chart line. This aims to help users avoid misinterpreting data.

The parameter manipulation factor indicates whether the users can filter out lines displayed. For example, when two lines are displayed, filtering can be applied to show only one line. This aims to help users simplify complex charts for easier interpretation.

Based on Table 3, comparing Recharts, Chart Js, Nivo, Plotly, and PSTrace, we can see that Plotly was most suitable for our needs. Plotly has advantages related to annotation (reference lines) and to interactivity (details and parameter manipulation). Compared with the other software, Plotly is very capable regarding these factors, therefore, Plotly was selected as the software framework for this project.

**CV result presentation.** Figs 14–17 present a visual comparison of cyclic voltammetry (CV) data rendered using four popular JavaScript charting libraries: Recharts, Chart.js, Nivo, and Plotly. These libraries were tested within the front-end interface of the proposed web-based heart disease detection system to evaluate their effectiveness in visualizing electrochemical sensor data. The purpose of this comparison was to assess which charting tool best supports the presentation and interpretation of voltammetry data—specifically for clinical and diagnostic use, in this case detecting the cardiac biomarker cardiac troponin. Each library's capability was evaluated based on three key criteria:

1. Annotation support, such as reference lines to mark peaks and baselines.

2. Interactivity support, including the ability to display tooltips or additional information when users hover over data points.

3. Parameter manipulation support, such as filtering data series for clarity.

The result demonstrated that Plotly offers the most comprehensive features, including interactive annotations and data filtering, making it the most suitable for accurate and user-friendly analysis of electrochemical voltammetry data. Thus, it was concluded that Plotly was the optimal choice for the front-end visualization framework for the proposed medical diagnostic system.

Recharts, as shown in Fig 14, supports basic annotation and hover interactivity but lacks parameter filtering features. While usable, the limited customization reduces its suitability for clinical diagnostic interpretation.

The Chart.js library, as shown in Fig 15, lacks built-in annotation and hover interactivity in its default configuration, which limits its effectiveness for identifying voltammetric peaks and baselines in biomedical applications.

**Table 3. Table of DPV data chart completeness test results.**

| No | Factors | Recharts | Chart Js | Nivo | Plotly | PSTrace |
|----|---------|----------|----------|------|--------|---------|
| 1. | Annotation – reference lines | Yes | No | No | Yes | No |
| 2. | Interactivity – details | Yes | No | Yes | Yes | No |
| 3. | Interactivity – parameter manipulation | No | No | No | Yes | No |

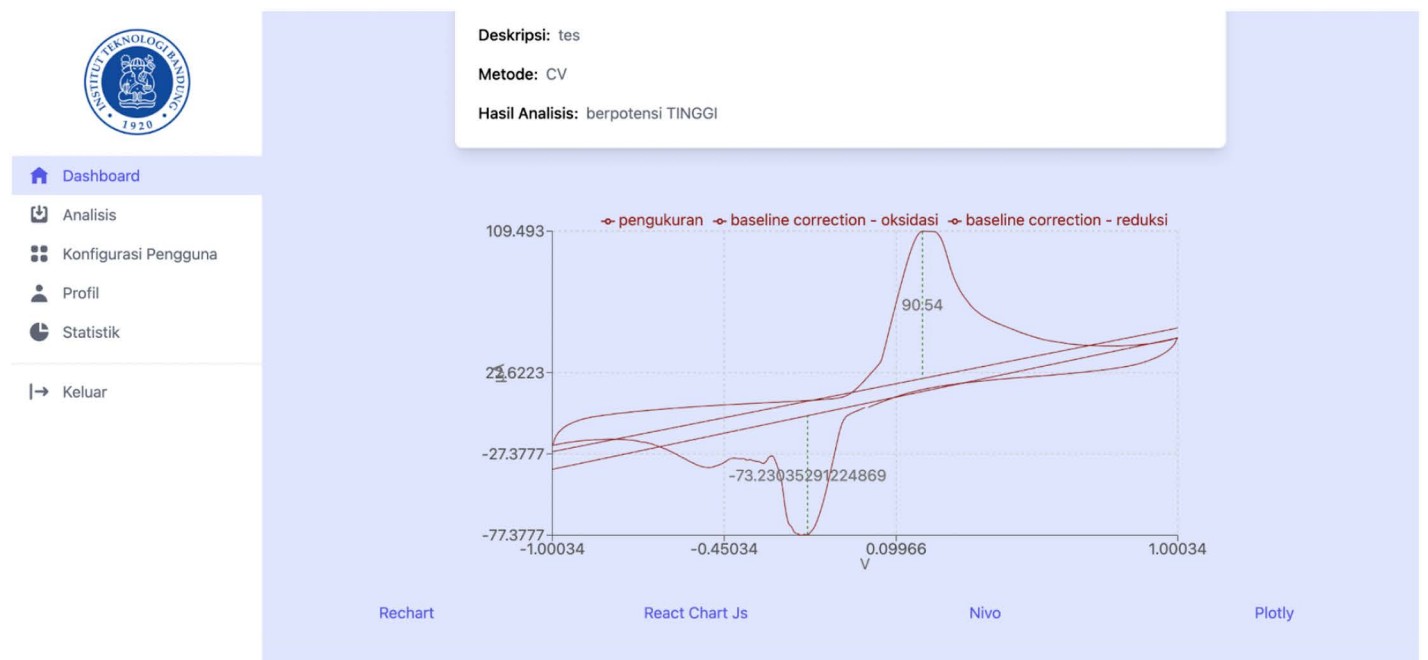

**Fig 14. Visualization of cyclic voltammetry (CV) data using Recharts.**

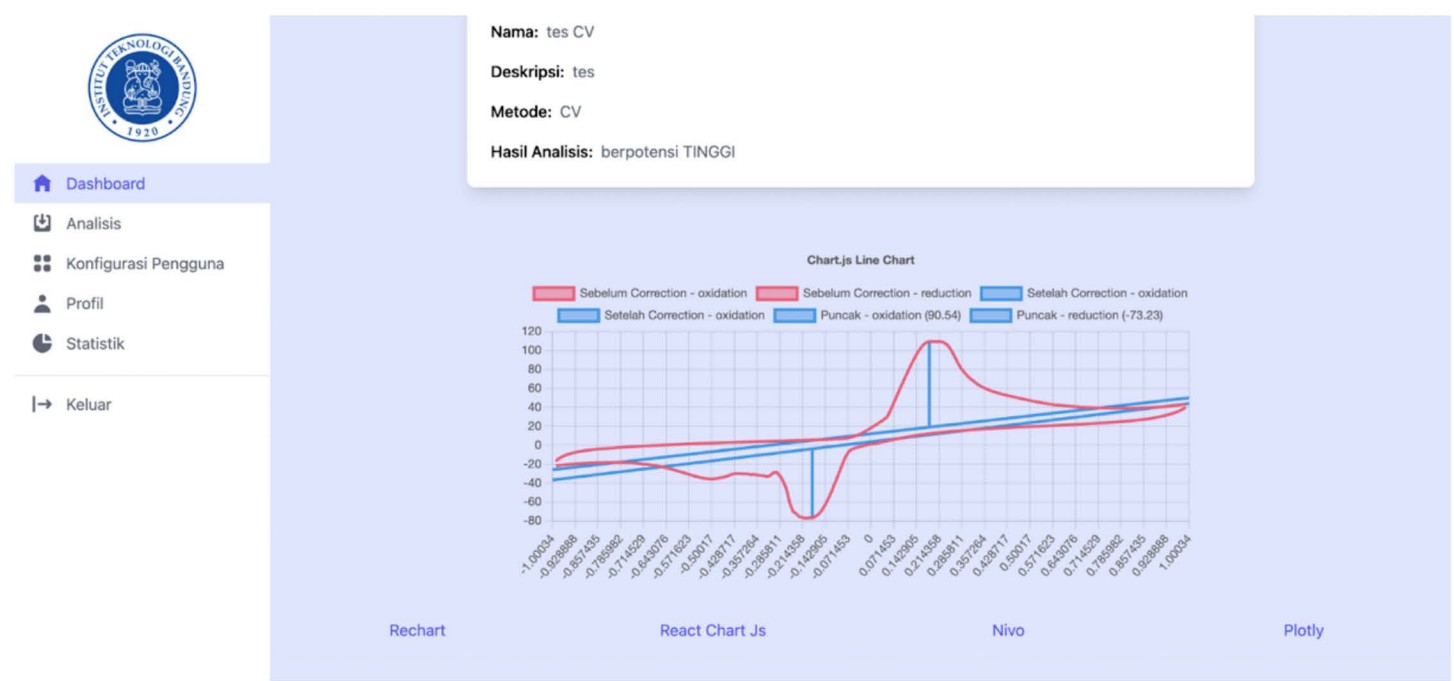

**Fig 15. Visualization of CV data using Chart.js.**

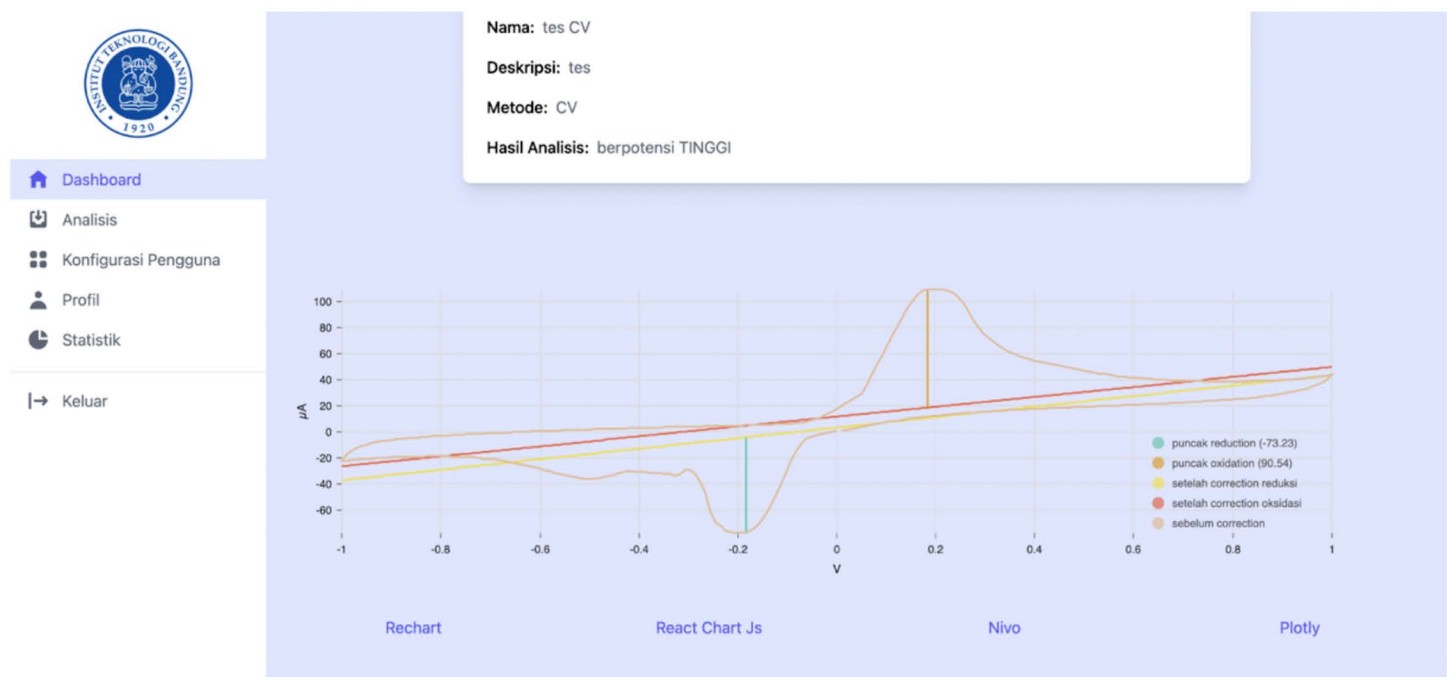

**Fig 16. Visualization of CV data using Nivo.**

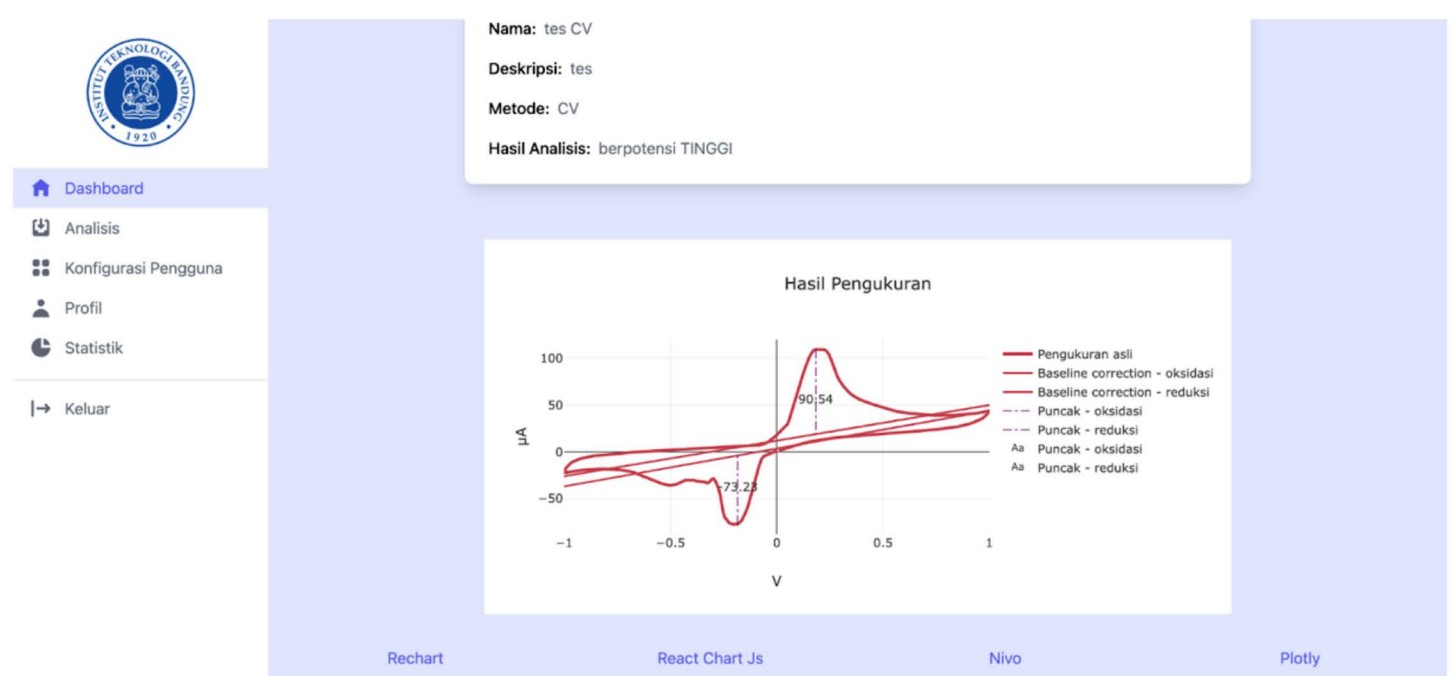

**Fig 17. Visualization of CV data using Plotly.**

The Nivo tool, as shown in Fig 16, offers improved interactivity over Chart.js, including tooltip support. However, it lacks annotation and filtering capabilities, making detailed analysis and result validation less intuitive for users.

Plotly, as shown in Fig 17, provides the most complete visualization features, including reference lines for annotation, interactive tooltips, and data filtering. These capabilities support precise identification of peaks and baselines, making Plotly the most suitable library for CV data visualization in clinical diagnostics.

**DPV result presentation.** Figs 18–21 compare differential pulse voltammetry (DPV) data visualizations using four different JavaScript charting libraries: Recharts, Chart.js, Nivo, and Plotly. Each library was integrated and tested within the front end of the web-based diagnostic system to assess its suitability for accurately representing electrochemical sensor measurements, particularly the current response peaks and baseline signals critical for heart disease detection.

Fig 18 shows Recharts, which demonstrates a clean visualization with support for annotations and basic interactivity. However, its limited ability to manipulate parameters restricts its usefulness in isolating or comparing specific data series, which is often necessary for distinguishing diagnostic patterns.

Fig 19 shows Chart.js, which has the most minimal feature set. It lacks annotation support, hover-based data interactivity, and filtering capabilities, making it the least suitable for diagnostic voltammetry applications, where precision and clarity are essential.

Fig 20 shows Nivo, which displays moderate performance with tooltip-based interactivity but still lacking essential features such as reference line annotations and parameter manipulation. This diminishes its capacity for in-depth interpretation of DPV graphs, especially when baseline and peak differentiation is required.

Fig 21 shows Plotly, which provides the most comprehensive and diagnostically valuable visualization. It supports all critical features: interactive tooltips, reference line annotations, and parameter filtering. These afford users—particularly lab technicians and clinicians—enhanced ability to identify, analyze, and interpret peak voltammetric signals, which are essential for evaluating cardiac troponin concentrations and predicting heart disease risk.

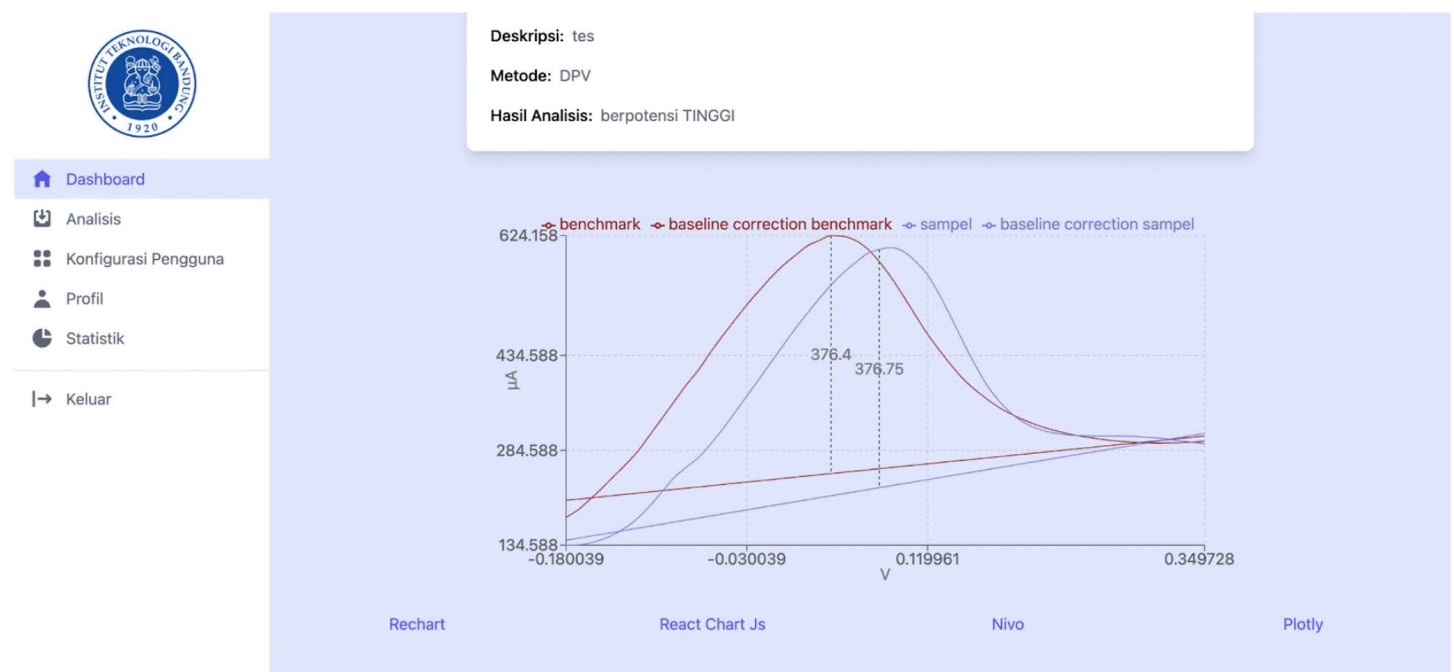

**Fig 18. Visualization of DPV data using Recharts.**

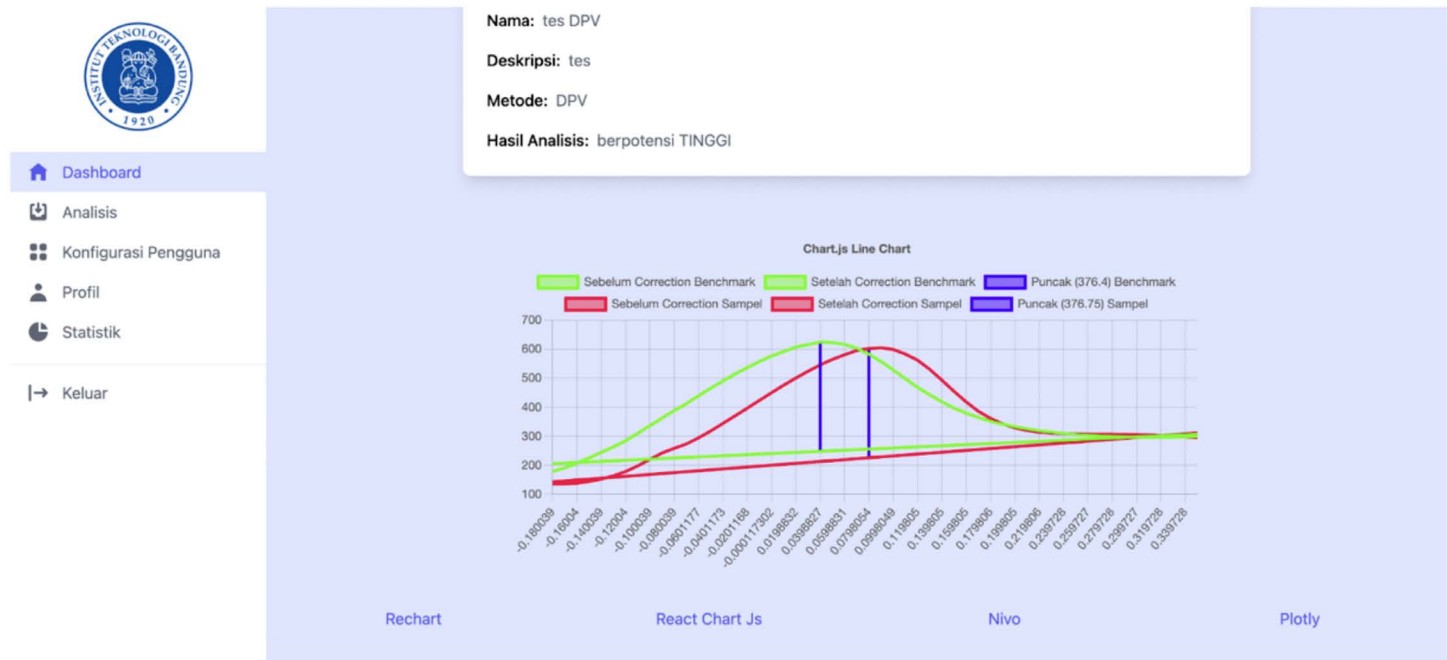

**Fig 19. Visualization of DPV data using Chart.js.**

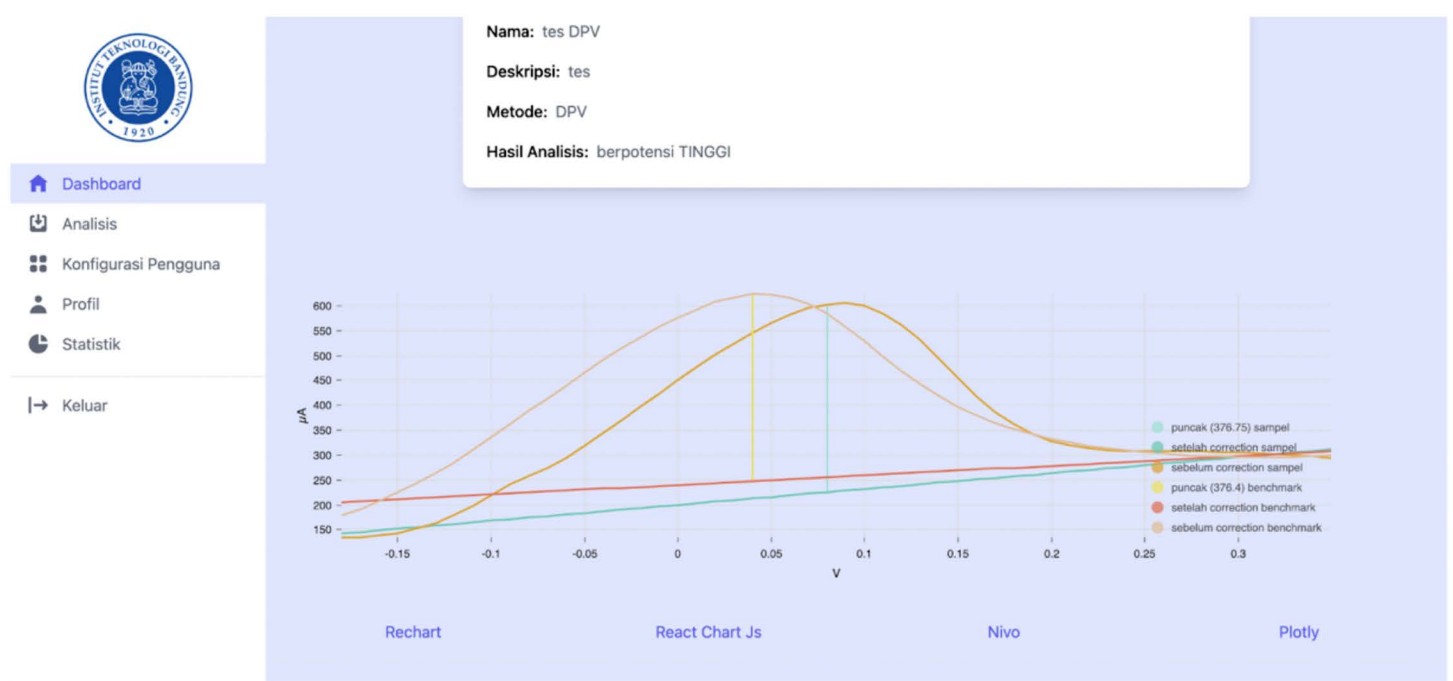

**Fig 20. Visualization of DPV data using Nivo.**

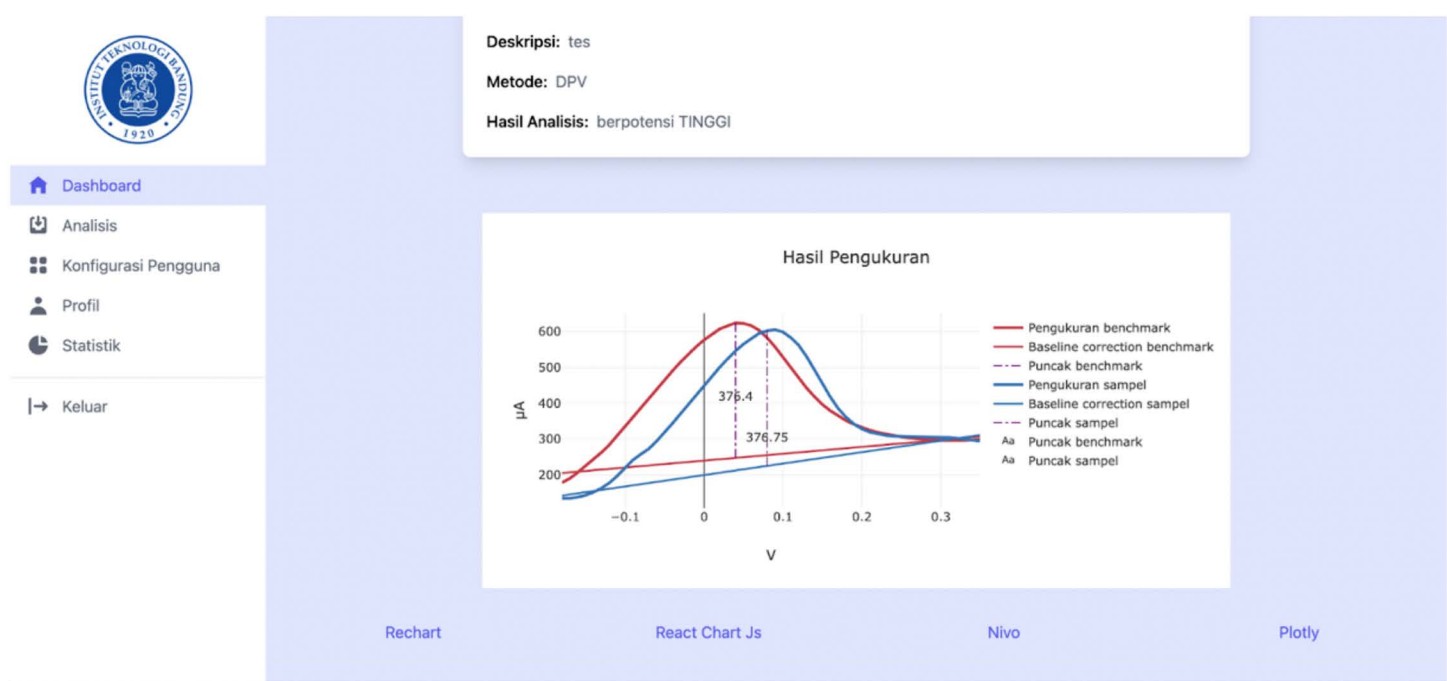

**Fig 21. Visualization of DPV data using Plotly.**

The comparison demonstrated that Plotly is the most appropriate visualization library for use in medical diagnostic applications involving voltammetric data, due to its completeness, interactivity, and support for data exploration tasks relevant to clinical analysis.

### Testing for data analysis

**Load testing.** Load testing was done to assess the speed, response time, reliability, and scalability of the system. This method identifies weak points and measures system performance under the load generated by multiple concurrent users. The tool used for this load testing was Locust. The load testing focused on two key application features that are frequently accessed by users: retrieving the list of measurements and opening the measurement detail page. The load testing was performed by simulating several concurrent users making requests to the server for these two key features, as shown in Table 4. The configuration for the load testing was as follows:

*Concurrent users*: 100 users, assuming the application is used by some special users such as lab technicians and some public users, as an initial test case.

*Testing duration*: 90 seconds, based on the average time users spend on a website, which is approximately 90 seconds.

The load testing on the back-end system demonstrated high throughput and consistent performance, with a total of 16,074 requests processed in 90 seconds. The average response times were 330 ms for retrieving the list of measurements and 410 ms for opening measurement detail pages, both maintaining a 100% success rate. The maximum response times were 1,280 ms and 1,314 ms, respectively, indicating the system's capability to handle peak loads without significant performance degradation. These results confirm that the back-end system is robust, reliable, and scalable, efficiently supporting concurrent user requests.

**Table 4. Testing configuration.**

| Request | Parameter | Value |
|---|---|---|
| GET/api/measurements | Total number of requests | 8,050 |
| | Average response time | 330 ms |
| | Success rate | 100% |
| | Maximum response time | 1,280 ms |
| GET/api/measurements/{id} | Total number of requests | 8,024 |
| | Average response time | 410 ms |
| | Success rate | 100% |
| | Maximum response time | 1,314 ms |

## Discussion

### Data visualization

Figs 22–25 provide a comparative overview of cyclic voltammetry (CV) and differential pulse voltammetry (DPV) outputs generated using the implemented web-based system versus those obtained from PSTrace, a widely recognized electrochemical analysis software. The aim of this comparison was to validate the accuracy and diagnostic reliability of the visualization and baseline correction algorithms integrated into the proposed system, specifically in the context of cardiac biomarker detection.

Each figure demonstrates the capability of the system to render voltammetric data in a clinically meaningful way by emphasizing two key diagnostic elements: the peak signal and the baseline. These elements are critical for assessing current responses in the context of cardiac troponin measurement.

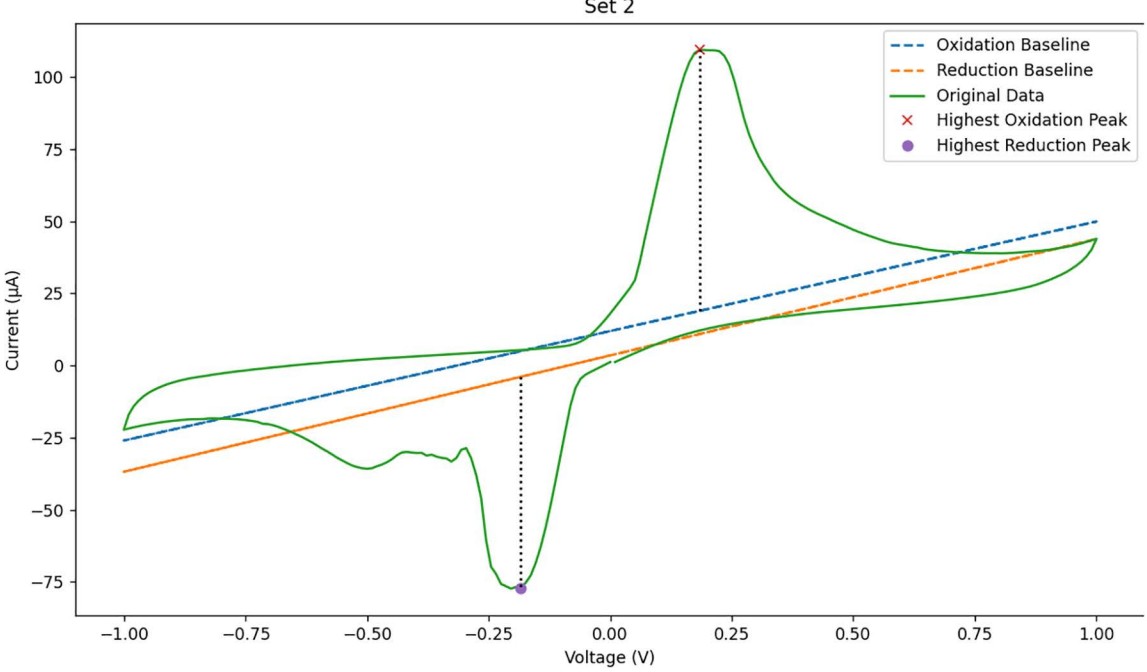

**Fig 22. CV result generated by the implemented system.**

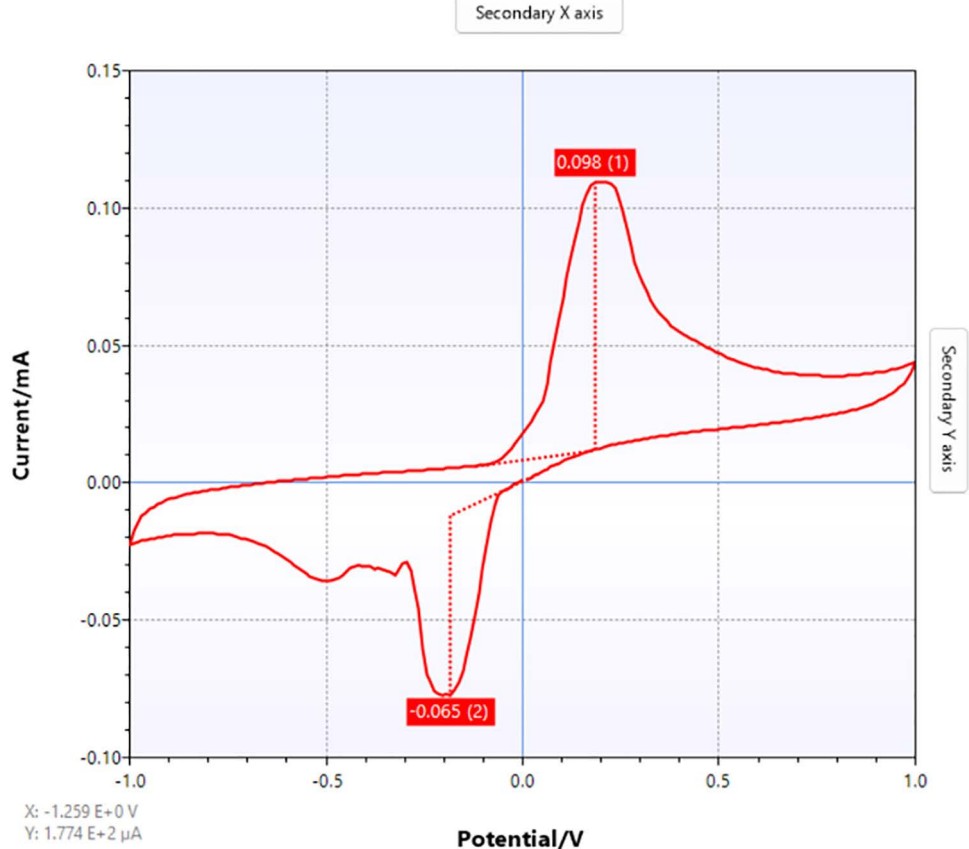

Selected curve: » CV i vs E Scan 1

X: -1.259 E+0 V
Y: 1.774 E+2 µA

**Fig 23. CV output using PSTrace.**

Fig 22 shows that the voltammogram displays multiple peaks with visible baseline shifts, indicating that while the key features are detectable, further refinement of the polynomial fitting algorithm may be necessary to achieve cleaner peak resolution, especially for redox peak pairs.

Fig 23 shows well-resolved oxidation and reduction peaks with minimal baseline distortion, providing a sharper and more accurate voltammogram. The baseline stability in PSTrace served as a benchmark for evaluating the visual quality of the CV rendering in the proposed system.

Fig 24 shows that a distinct single peak is clearly visible, with the baseline appropriately corrected using the ALS algorithm. The peak's visibility and shape demonstrate that the implemented system is effective for DPV data analysis, supporting accurate detection of a low-concentration analyte such as cardiac troponin.

Fig 25 illustrates a similarly sharp and stable single peak with a well-defined baseline. The similarity to Fig 24 confirms that the system's implementation of ALS-based correction and peak detection is in close alignment with established analytical software.

## Analysis visualization

The cyclic voltammetry (CV) analysis algorithm was developed based on the methodology described in the corresponding subsection. For validation, identical datasets were processed using both the proposed algorithm and the reference

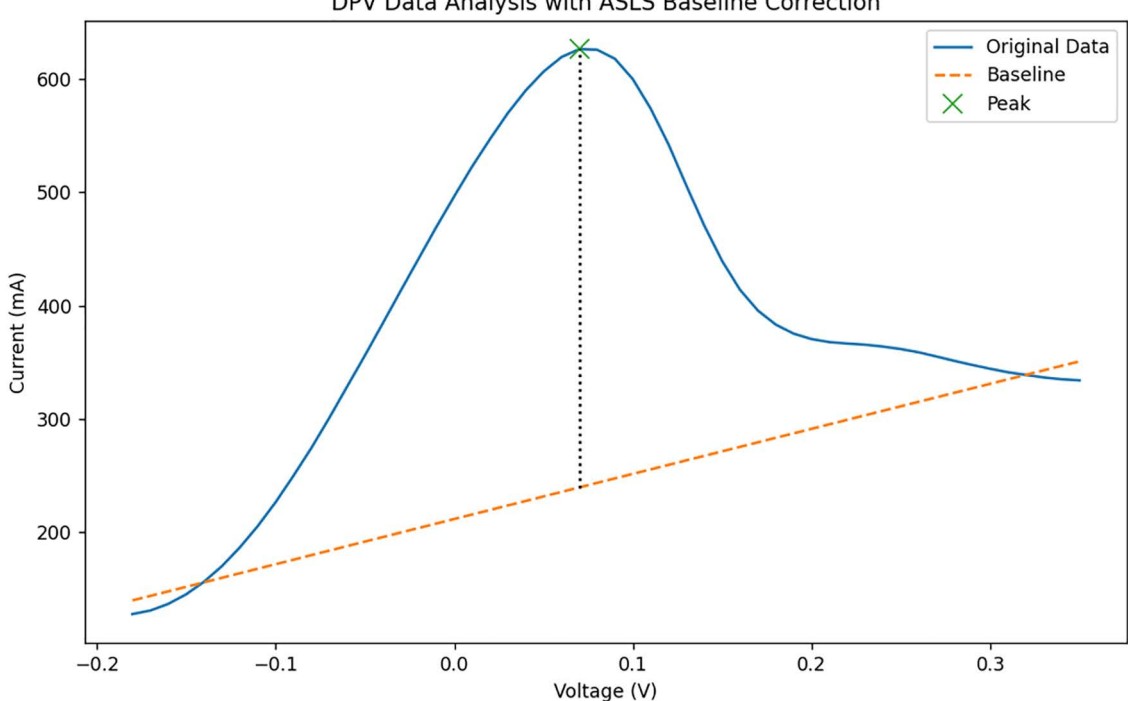

**Fig 24. DPV result generated by the implemented system.**

software, PSTrace. The comparison indicates that the baseline estimated by PSTrace demonstrates higher accuracy and stability compared to that produced by the implemented CV algorithm, particularly in handling baseline drift.

For differential pulse voltammetry (DPV), the same evaluation approach was applied using identical datasets. The baseline generated by the proposed DPV algorithm, based on asymmetric least squares (ALS), shows close agreement with the baseline obtained from PSTrace. This consistency suggests that the implemented DPV approach is reliable for baseline estimation and peak interpretation.

Accurate baseline estimation is essential in voltammetric analysis, as it directly affects the extraction of peak-to-baseline current responses, which serve as key indicators of electrochemical activity associated with cardiac biomarkers. Therefore, the observed agreement in DPV baseline estimation supports the robustness of the proposed method for interpreting electrochemical signals.

### Chart

Based on the comparison presented in Table 3, Plotly provides the most comprehensive set of features required for the proposed system, particularly in terms of annotation, interactivity, and parameter manipulation. Compared to other libraries such as Recharts, Chart.js, and Nivo, Plotly offers superior capabilities for visualizing voltammetric data in a clear and interpretable manner.

Furthermore, when benchmarked against PSTrace, Plotly demonstrates comparable—and in some aspects enhanced—functionality, especially in supporting interactive exploration and dynamic visualization of electrochemical signals. These features are essential for accurately identifying baseline levels and peak characteristics in voltammetry data.

Therefore, Plotly was selected as the primary charting library due to its completeness, flexibility, and suitability for supporting analytical and diagnostic visualization within the proposed system.

Selected curve: » DPV i vs E

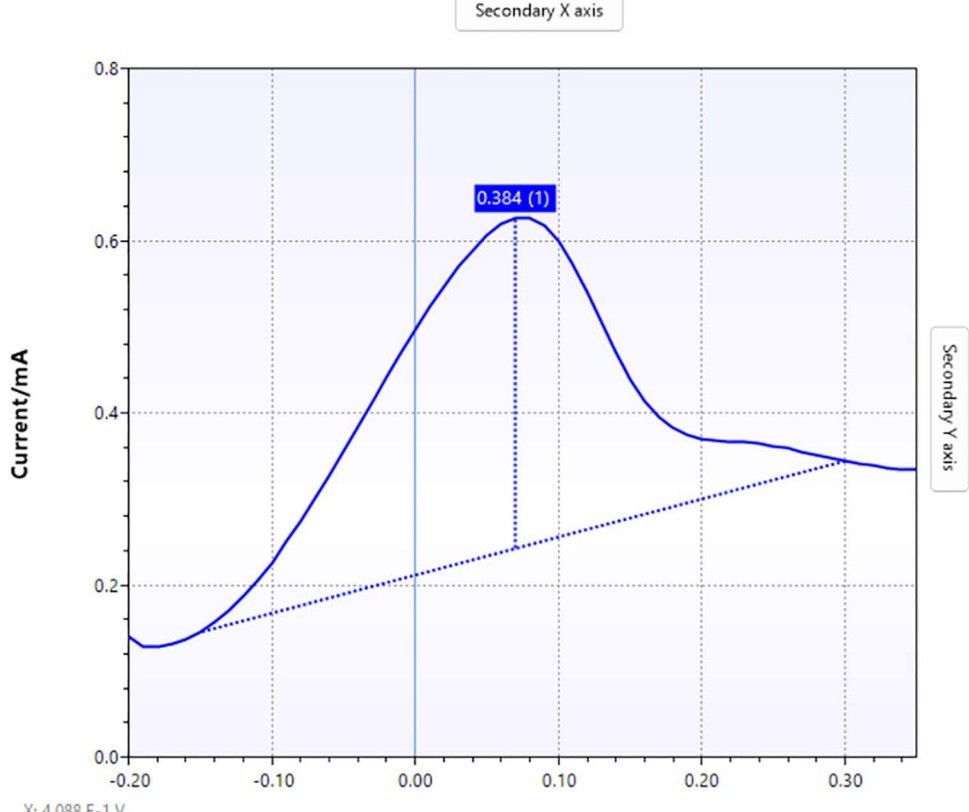

Fig 25. DPV output using PSTrace.

## Conclusion

This study presents the development of a web-based system that transforms raw voltammetry data into interpretable diagnostic information for early detection of cardiovascular conditions. By integrating electrochemical sensing techniques with cloud-based data management and interactive visualization, the system enables reliable estimation of cardiac biomarker responses, particularly cardiac troponin.

The proposed framework combines automated signal-processing methods—including baseline correction, peak detection, and current-response extraction—with a scalable web architecture. The back-end implementation using FastAPI and PostgreSQL demonstrates robust performance in handling large volumes of electrochemical data, while the React-based front end with Plotly visualization provides an intuitive and interactive interface accessible to both clinical practitioners and non-expert users.

The results highlight the effectiveness of algorithm-assisted interpretation in reducing noise, minimizing operator-dependent variability, and improving the reproducibility of voltammetric analysis. By converting complex electrochemical signals into structured and interpretable outputs, the system addresses a critical gap between biosensor data acquisition and actionable clinical insight. In addition, the platform supports remote data access and monitoring, which is essential for improving early screening and preventive healthcare strategies.

Future work will focus on enhancing the accuracy of voltammetric signal interpretation through advanced baseline correction approaches, including adaptive and machine learning-based methods. Furthermore, integration with real-time data streams from mobile or wearable biosensors will enable continuous cardiac monitoring in home-care settings. The incorporation of electronic health record (EHR) systems and clinical decision-support modules is also expected to further improve the system's applicability, scalability, and impact in broader healthcare environments.

## Author contributions

**Conceptualization:** Wikan Danar Sunindyo, Isa Anshori.

**Funding acquisition:** Wikan Danar Sunindyo.

**Methodology:** Isa Anshori.

**Supervision:** Wikan Danar Sunindyo, Infall Syafalni, Latifa Dwiyanti.

**Writing – original draft:** Kristo Abdi Wiguna, Marcelus Michael Herman Kahari, Uperianti.

**Writing – review & editing:** Kristo Abdi Wiguna, Marcelus Michael Herman Kahari, Uperianti.

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
