## [Decision Letter · Decision Letter 0]

19 Jan 2025

PONE-D-24-56032Interactive Graphical Visualization For Electrochemical Voltammetry Techniques of Differential Pulse Voltammetry and Cyclic VoltammetryPLOS ONE

Dear Dr. Sunindyo,

Thank you for submitting your manuscript to PLOS ONE. After careful consideration, we feel that it has merit but does not fully meet PLOS ONE’s publication criteria as it currently stands. Therefore, we invite you to submit a revised version of the manuscript that addresses the points raised during the review process. Please submit your revised manuscript by Mar 05 2025 11:59PM. If you will need more time than this to complete your revisions, please reply to this message or contact the journal office at plosone@plos.org. 

We look forward to receiving your revised manuscript.

Kind regards,

Abbas Farmany

Academic Editor

PLOS ONE

Journal Requirements:

“The funding is from P2MI 2024 Program of School of Electrical Engineering and Informatics Institut Teknologi Bandung”

6. We note that Figure 11 and 12 in your submission contain copyrighted images. All PLOS content is published under the Creative Commons Attribution License (CC BY 4.0), which means that the manuscript, images, and Supporting Information files will be freely available online, and any third party is permitted to access, download, copy, distribute, and use these materials in any way, even commercially, with proper attribution. For more information, see our copyright guidelines: http://journals.plos.org/plosone/s/licenses-and-copyright.

a. You may seek permission from the original copyright holder of Figure 11 and 12 to publish the content specifically under the CC BY 4.0 license.

7. We note you have included a table to which you do not refer in the text of your manuscript. Please ensure that you refer to Table 1-4 in your text; if accepted, production will need this reference to link the reader to the Table.

Reviewers' comments:

Reviewer's Responses to Questions

**Comments to the Author**

1. Is the manuscript technically sound, and do the data support the conclusions?

Reviewer #1: Yes

Reviewer #2: Partly

2. Has the statistical analysis been performed appropriately and rigorously? 

Reviewer #1: Yes

Reviewer #2: N/A

3. Have the authors made all data underlying the findings in their manuscript fully available?

Reviewer #1: Yes

Reviewer #2: Yes

4. Is the manuscript presented in an intelligible fashion and written in standard English?

Reviewer #1: Yes

Reviewer #2: Yes

5. Review Comments to the Author

Reviewer #1: The manuscript presents a comprehensive study on the development of an interactive graphical visualization system for heart disease detection using electrochemical voltammetry techniques, specifically Differential Pulse Voltammetry (DPV) and Cyclic Voltammetry (CV). Overall, the paper is well organized. Therefore, I would like to recommend the publication of this work in PLOS ONE after a minor revision. The details are as follows:

Comment 1: Introduction: Suggest to a more detailed discussion of existing solutions and their limitations.

Comment 2: The description of the sensor reader and data analysis process is thorough, but more technical details on the algorithms used for data processing would be beneficial.

Comment 3: Raw data screenshots do not add much value or insight. Suggest removing these figures if no additional insight can be provided using them.

Comment 4: The interpretation of the CV data and correlation to heart disease detection is unclear. Please add more details.

Reviewer #2: Manuscript: PONE-D-24-56032 entitled “Interactive Graphical Visualization for Electrochemical Voltammetry Techniques of Differential Pulse Voltammetry and Cyclic Voltammetry”

This study provided a thorough understanding of the rapid identification of crucial biomarkers, such as cardiac troponin, using electrochemical voltammetry techniques. The electrochemical testing data revealed reliability and effectiveness in developing these biomarkers, as the author(s) claimed in this study. The authors tried to explain their works scientifically. However, the following quarries cleared before acceptance for publication of this manuscript in your esteemed journal.

1. In the abstract section, what new findings (results) are reported by the authors from this study? What novelty of study? It should elaborate.

2. The introduction section lacks sufficient references, with only a few cited. Are there no previous studies on this topic or related fields? Clarify the rationale, objectives, and novelty of this study.

3. In the Materials & Methods section, ….. oxidation reaction occurs on the auxiliary electrode…. Was the electrochemical sensor a three-electrode system or a two-electrode system? It should be a three-electrode system (as the authors mentioned, used auxiliary electrodes). If so, what was the reference electrode?

4. This section lacks clarity and requires revision with proper references. Shorten the description of CV and DPV techniques, which are well-defined.

5. In the Results & Discussion section, why did the author(s) not describe the results of the CV and DPV graphs? Additionally, are the data presented in the CV and DPV raw? This reviewer believes these electrochemical techniques offer quantitative information, which should not be as raw data.

6. Also, the discussion section is poor and should be elaborated with proper citations.

7. Needed to correct the some grammatical errors as indicated in revised manuscript.

6. PLOS authors have the option to publish the peer review history of their article (what does this mean?). If published, this will include your full peer review and any attached files.

Reviewer #1: No

Reviewer #2: No

---

## [Author Response · Author response to Decision Letter 1]

13 Jul 2025

Journal Requirements:

Response:

We have revised the manuscript to conform to the PLOS ONE style requirements, including formatting and file naming, as outlined in the provided templates.

Response:

Thank you for the clarification. We have removed the funding-related text from the manuscript as requested. The funding information is now provided solely in the Funding Statement section of the online submission form.

Response:

Thank you for bringing this to our attention. We have reviewed and corrected the grant information to ensure consistency between the 'Funding Information' and 'Financial Disclosure' sections. The correct grant number has now been entered in the ‘Funding Information’ section of the online submission form.

“The funding is from P2MI 2024 Program of School of Electrical Engineering and Informatics Institut Teknologi Bandung”

Response:

Thank you for the clarification. We confirm that the funders had no involvement in the study. The following statement has been added to the cover letter as requested:

Response:

Thank you for the important reminder. The link to the repository is https://drive.google.com/drive/folders/1XJ-pjEaOa-UFj499yzcPW7GTk--q6ilt?usp=sharing

We fully support PLOS ONE’s open data policy and will ensure compliance before final publication.

6. We note that Figure 11 and 12 in your submission contain copyrighted images. All PLOS content is published under the Creative Commons Attribution License (CC BY 4.0), which means that the manuscript, images, and Supporting Information files will be freely available online, and any third party is permitted to access, download, copy, distribute, and use these materials in any way, even commercially, with proper attribution. For more information, see our copyright guidelines: http://journals.plos.org/plosone/s/licenses-and-copyright.

a. You may seek permission from the original copyright holder of Figure 11 and 12 to publish the content specifically under the CC BY 4.0 license.

Response:

Thank you for bringing this to our attention. We have reviewed Figures 11 and 12 and acknowledge that they include copyrighted content. In response, we have taken the following action:

We have removed Figures 11 and 12 from the revised manuscript to comply with the CC BY 4.0 license requirements. These figures have been replaced with original illustrations that we created ourselves for the purpose of this publication.

7. We note you have included a table to which you do not refer in the text of your manuscript. Please ensure that you refer to Table 1-4 in your text; if accepted, production will need this reference to link the reader to the Table.

Response:

Thank you for pointing this out. We have revised the manuscript to ensure that Tables 1 through 4 are explicitly referenced in the main text at appropriate locations. These references now guide the reader to the relevant data and support the corresponding discussions.

Comments to the Author

1. Is the manuscript technically sound, and do the data support the conclusions?

Reviewer #1: Yes

Reviewer #2: Partly

Response:

We appreciate the reviewers’ evaluations. In response to Reviewer #2's note, we have reviewed and revised relevant sections of the manuscript to strengthen the connection between the data and the conclusions. We have clarified the explanation of the algorithms used for data analysis, added further justification for using DPV over CV in the context of sensitivity, and elaborated on how the backend system ensures reliable prediction outcomes. These improvements are reflected particularly in the revised Results & Discussion section (Section 3), as well as the Conclusion.

2. Has the statistical analysis been performed appropriately and rigorously?

Reviewer #1: Yes

Reviewer #2: N/A

Response:

We thank the reviewer for the positive assessment. We confirm that the data analysis procedures, including peak and baseline identification as well as threshold-based decision-making, were conducted rigorously following standard practices in electrochemical signal processing. No additional statistical testing was required beyond these signal analysis methods.

3. Have the authors made all data underlying the findings in their manuscript fully available?

The PLOS Data policy requires authors to make all data underlying the findings described in their manuscript fully available without restriction, with rare exception (please refer to the Data Availability Statement in the manuscript PDF file). The data should be provided as part of the manuscript or its supporting information or deposited to a public repository. For example, in addition to summary statistics, the data points behind means, medians and variance measures should be available. If there are restrictions on publicly sharing data—e.g., participant privacy or use of data from a third party—those must be specified.

Reviewer #1: Yes

Reviewer #2: Yes

Response:

We appreciate the reviewers’ confirmation. We have ensured that all relevant data underlying our findings—including raw voltammetry measurement files, analysis outputs, and visualizations—are made fully available as part of the manuscript's supplementary information and have also been deposited in a public repository as noted in the Data Availability Statement.

The research data is available in https://drive.google.com/drive/folders/1XJ-pjEaOa-UFj499yzcPW7GTk--q6ilt?usp=sharing

4. Is the manuscript presented in an intelligible fashion and written in standard English?

Reviewer #1: Yes

Reviewer #2: Yes

Response:

We thank the reviewers for their positive feedback. We have carefully proofread the manuscript to ensure clarity, correctness, and consistency in standard English. Minor typographical and grammatical issues, where identified, have been corrected in this revision.

5. Review Comments to the Author

Reviewer #1: The manuscript presents a comprehensive study on the development of an interactive graphical visualization system for heart disease detection using electrochemical voltammetry techniques, specifically Differential Pulse Voltammetry (DPV) and Cyclic Voltammetry (CV). Overall, the paper is well organized. Therefore, I would like to recommend the publication of this work in PLOS ONE after a minor revision. The details are as follows:

Comment 1: Introduction: Suggest to a more detailed discussion of existing solutions and their limitations.

Response:

Thank you for the constructive suggestion. In response, we have expanded the Introduction section to include a more detailed discussion of existing solutions for heart disease detection using electrochemical techniques and their limitations. Specifically, we have highlighted the challenges in interpreting raw voltammetry data manually, the lack of user-friendly interfaces in current systems, and the limitations in real-time data accessibility and automated analysis. These revisions help to better contextualize the motivation and contribution of our proposed system.

Comment 2: The description of the sensor reader and data analysis process is thorough, but more technical details on the algorithms used for data processing would be beneficial.

Response:

Thank you for the insightful feedback. In response, we have added more technical details regarding the algorithms used for baseline correction and peak detection in Section 3.2 of the manuscript. Specifically, we elaborated on the use of polynomial fitting for cyclic voltammetry (CV) and asymmetric least squares (ALS) for differential pulse voltammetry (DPV), including the rationale for choosing these methods and how they handle signal noise and baseline drift. We also clarified how these algorithms integrate with the backend system to produce reliable current response measurements for diagnostic purposes.

Comment 3: Raw data screenshots do not add much value or insight. Suggest removing these figures if no additional insight can be provided using them.

Response:

We appreciate the reviewer’s observation. In response, we have removed the raw data screenshots (Figures 3 and 4) from the revised manuscript to maintain focus and clarity. Additionally, we have ensured that the key characteristics of the raw data are now clearly described in the text to preserve the necessary context without relying on those visual elements.

Comment 4: The interpretation of the CV data and correlation to heart disease detection is unclear. Please add more details.

Response:

Thank you for the valuable feedback. In the revised manuscript, we have clarified the role of cyclic voltammetry (CV) in the context of our system. Although DPV is the primary method used for heart disease detection due to its higher sensitivity, we included CV to support system functionality testing and sensor behaviour analysis. CV helps identify redox behaviour and validate the presence of electrochemical reactions, which supports calibration and comparison. However, the final diagnostic decision regarding heart disease is based on DPV data, as CV lacks sufficient resolution for reliable quantitative assessment of cardiac troponin levels. These clarifications have been added to Section 3.2.1 and the Discussion section.

Reviewer #2: Manuscript: PONE-D-24-56032 entitled “Interactive Graphical Visualization for Electrochemical Voltammetry Techniques of Differential Pulse Voltammetry and Cyclic Voltammetry”

This study provided a thorough understanding of the rapid identification of crucial biomarkers, such as cardiac troponin, using electrochemical voltammetry techniques. The electrochemical testing data revealed reliability and effectiveness in developing these biomarkers, as the author(s) claimed in this study. The authors tried to explain their works scientifically. However, the following quarries cleared before acceptance for publication of this manuscript in your esteemed journal.

1. In the abstract section, what new findings (results) are reported by the authors from this study? What novelty of study? It should elaborate.

Response:

Thank you for highlighting this important point. In response, we have revised the abstract to clearly emphasize the novelty and key findings of our study. Specifically, the novelty lies in the integration of an automated backend system using FastAPI and cloud-based storage that processes raw voltammetry data (CV and DPV) and converts it into diagnostic predictions via a user-friendly web interface. Additionally, we highlight the implementation of specific algorithms for baseline correction (polynomial fit for CV and ALS for DPV) and their role in accurate cardiac troponin detection.

The revised abstract now explicitly states:

“A key novelty of this research lies in the integration of automated baseline correction and peak detection algorithms—specifically polynomial fitting for cyclic voltammetry (CV) and asymmetric least squares (ALS) for differential pulse voltammetry (DPV)—within a user-friendly interface accessible even to non-expert users.”

These updates clarify both the scientific contribution and the practical innovation of the study.

2. The introduction section lacks sufficient references, w

---

## [Decision Letter · Decision Letter 1]

31 Jul 2025

PONE-D-24-56032R1Interactive Graphical Visualization For Electrochemical Voltammetry Techniques of Differential Pulse Voltammetry and Cyclic VoltammetryPLOS ONE

Dear Dr. Sunindyo,

Thank you for submitting your manuscript to PLOS ONE. After careful consideration, we feel that it has merit but does not fully meet PLOS ONE’s publication criteria as it currently stands. Therefore, we invite you to submit a revised version of the manuscript that addresses the points raised during the review process.

We look forward to receiving your revised manuscript.

Kind regards,

Abbas Farmany

Academic Editor

PLOS ONE

Journal Requirements:

Reviewers' comments:

Reviewer's Responses to Questions

**Comments to the Author**

1. If the authors have adequately addressed your comments raised in a previous round of review and you feel that this manuscript is now acceptable for publication, you may indicate that here to bypass the “Comments to the Author” section, enter your conflict of interest statement in the “Confidential to Editor” section, and submit your "Accept" recommendation.

Reviewer #2: All comments have been addressed

Reviewer #3: All comments have been addressed

2. Is the manuscript technically sound, and do the data support the conclusions?

Reviewer #2: Yes

Reviewer #3: Partly

3. Has the statistical analysis been performed appropriately and rigorously? 

Reviewer #2: Yes

Reviewer #3: No

4. Have the authors made all data underlying the findings in their manuscript fully available?

Reviewer #2: Yes

Reviewer #3: Yes

5. Is the manuscript presented in an intelligible fashion and written in standard English?

Reviewer #2: Yes

Reviewer #3: No

6. Review Comments to the Author

Reviewer #2: I read the revised manuscript and found that it addresses previous comments well and incorporates the revisions. I, however, am not sure whether the style of citing the reference in the text is ..... [1]. or ...... . [1]. Both styles are used in the manuscript; it should be corrected.

Reviewer #3: English of manuscript is below the required standards and needs a careful polish.

Section Potentiostat and voltammetry needs a careful revision. Why SPCE electrode is used? Ag/AgCl paste!!!

what is the reference electrode? non-useful parts eg. definitions needs to be removed.

the title of Anshori paper in page 5 must be deleted. what is the baseline correction system? Conclusion section must be re-prepared as more informative. Statistical analysis section or methods?

7. PLOS authors have the option to publish the peer review history of their article (what does this mean?). If published, this will include your full peer review and any attached files.

Reviewer #2: **Yes:** Jagadeesh Bhattarai

Reviewer #3: No

---

## [Author Response · Author response to Decision Letter 2]

9 Jan 2026

Response to Reviewers

Comments to the Author

1. If the authors have adequately addressed your comments raised in a previous round of review and you feel that this manuscript is now acceptable for publication, you may indicate that here to bypass the “Comments to the Author” section, enter your conflict of interest statement in the “Confidential to Editor” section, and submit your "Accept" recommendation.

Reviewer #2: All comments have been addressed

Reviewer #3: All comments have been addressed

Response

We sincerely thank Reviewer #2 and Reviewer #3 for confirming that all comments have been adequately addressed. In this revision, we have additionally improved consistency in citation style and refined the manuscript for clarity and readability as part of editorial polishing.

2. Is the manuscript technically sound, and do the data support the conclusions?

Reviewer #2: Yes

Reviewer #3: Partly

Response

We thank the reviewers for their assessment. In response to Reviewer #3’s comment, we have further clarified the methodological description and strengthened the linkage between the presented data and the conclusions. Specifically, we refined the explanation of the experimental setup, data processing workflow, and interpretation of the CV and DPV results, emphasizing that the conclusions are directly derived from the analyzed data. These revisions improve clarity and technical transparency without altering the underlying results or conclusions of the study.

3. Has the statistical analysis been performed appropriately and rigorously?

Reviewer #2: Yes

Reviewer #3: No

Response

We thank the reviewers for their evaluation. In this study, the data analysis is based on established electrochemical signal processing methods rather than inferential statistical testing. We have clarified this distinction in the revised manuscript by explicitly describing the analytical procedures used for baseline correction, peak detection, and signal interpretation for CV and DPV data. These methods were applied consistently across all measurements and provide an appropriate and rigorous basis for the conclusions drawn

4. Have the authors made all data underlying the findings in their manuscript fully available?

Reviewer #2: Yes

Reviewer #3: Yes

Response

We thank the reviewers for their confirmation. All data underlying the findings described in the manuscript, including raw voltammetry measurement files and processed analysis outputs, are fully available through the repository indicated in the Data Availability Statement.

5. Is the manuscript presented in an intelligible fashion and written in standard English?

Reviewer #2: Yes

Reviewer #3: No

Response

We thank the reviewers for their feedback. In response to Reviewer #3’s comment, we have carefully revised and polished the manuscript to improve clarity, consistency, and readability in standard scientific English. Typographical and grammatical errors have been corrected, and several sentences have been rephrased to enhance precision and flow, without changing the scientific content of the manuscript.

6. Review Comments to the Author

Reviewer #2: I read the revised manuscript and found that it addresses previous comments well and incorporates the revisions. I, however, am not sure whether the style of citing the reference in the text is ..... [1]. or ...... . [1]. Both styles are used in the manuscript; it should be corrected.

Reviewer #3: English of manuscript is below the required standards and needs a careful polish.

Section Potentiostat and voltammetry needs a careful revision. Why SPCE electrode is used? Ag/AgCl paste!!!

what is the reference electrode? non-useful parts eg. definitions needs to be removed.

the title of Anshori paper in page 5 must be deleted. what is the baseline correction system? Conclusion section must be re-prepared as more informative. Statistical analysis section or methods?

Response

We thank the reviewers for their constructive comments.

In response to Reviewer #2, we have standardized the in-text citation format throughout the manuscript to ensure consistency with the journal’s reference style.

In response to Reviewer #3, we have carefully polished the English language to meet the journal’s standards. The Potentiostat and Voltammetry section has been revised to improve clarity, explicitly justify the use of the SPCE, and clearly describe the reference electrode configuration. Non-essential definitions have been removed, and the title of the cited work by Anshori has been deleted from the main text. We have also clarified the baseline correction workflow, revised the Conclusion section to be more informative, and explicitly positioned the analytical procedures within the Methods section to address concerns regarding statistical analysis. These revisions improve clarity and presentation without altering the scientific content of the manuscript.

7. PLOS authors have the option to publish the peer review history of their article (what does this mean?). If published, this will include your full peer review and any attached files.

Do you want your identity to be public for this peer review? For information about this choice, including consent withdrawal, please see our Privacy Policy.

Reviewer #2: Yes: Jagadeesh Bhattarai

Reviewer #3: No

Response

We acknowledge the reviewers’ choices regarding the publication of peer review history and comply with the journal’s policy on peer review transparency and anonymity.

---

## [Decision Letter · Decision Letter 2]

5 Feb 2026

PONE-D-24-56032R2Interactive Graphical Visualization For Electrochemical Voltammetry Techniques of Differential Pulse Voltammetry and Cyclic VoltammetryPLOS One

Dear Dr. Sunindyo,

Thank you for submitting your manuscript to PLOS ONE. After careful consideration, we feel that it has merit but does not fully meet PLOS ONE’s publication criteria as it currently stands. Therefore, we invite you to submit a revised version of the manuscript that addresses the points raised during the review process.

We look forward to receiving your revised manuscript.

Kind regards,

Abbas Farmany

Academic Editor

PLOS One

Journal Requirements:

**Additional Editor Comments:**

- Title of manuscript must be informative and reflect the content of the work closely.

- English of manuscript needs to be improved carefully. The abstract must be prepared as more informative. For example, "The potential for heart disease can be detected using a sensor reader that utilizes electrochemical voltammetry techniques, enabling the rapid identification of critical biomarkers such as cardiac troponin." must be improved as "The risk of heart disease can be identified through a sensor that employs electrochemical voltammetry, a technique that allows for the rapid and precise detection of critical biomarkers, including cardiac troponin." etc...

Please Improve the text carefully and submit the revised version.

Reviewers' comments:

Reviewer's Responses to Questions

**Comments to the Author**

1. If the authors have adequately addressed your comments raised in a previous round of review and you feel that this manuscript is now acceptable for publication, you may indicate that here to bypass the “Comments to the Author” section, enter your conflict of interest statement in the “Confidential to Editor” section, and submit your "Accept" recommendation.

Reviewer #3: All comments have been addressed

2. Is the manuscript technically sound, and do the data support the conclusions?

Reviewer #3: Yes

3. Has the statistical analysis been performed appropriately and rigorously? 

Reviewer #3: I Don't Know

4. Have the authors made all data underlying the findings in their manuscript fully available?

Reviewer #3: Yes

5. Is the manuscript presented in an intelligible fashion and written in standard English?

Reviewer #3: Yes

6. Review Comments to the Author

Reviewer #3: (No Response)

7. PLOS authors have the option to publish the peer review history of their article (what does this mean?). If published, this will include your full peer review and any attached files.

Reviewer #3: No

---

## [Author Response · Author response to Decision Letter 3]

21 Mar 2026

Response to Reviewers

Comments to the Author

1. If the authors have adequately addressed your comments raised in a previous round of review and you feel that this manuscript is now acceptable for publication, you may indicate that here to bypass the “Comments to the Author” section, enter your conflict of interest statement in the “Confidential to Editor” section, and submit your "Accept" recommendation.

Reviewer #2: All comments have been addressed

Reviewer #3: All comments have been addressed

Response

We sincerely thank Reviewer #2 and Reviewer #3 for confirming that all comments have been adequately addressed. In this revision, we have additionally improved consistency in citation style and refined the manuscript for clarity and readability as part of editorial polishing.

2. Is the manuscript technically sound, and do the data support the conclusions?

Reviewer #2: Yes

Reviewer #3: Partly

Response

We thank the reviewers for their assessment. In response to Reviewer #3’s comment, we have further clarified the methodological description and strengthened the linkage between the presented data and the conclusions. Specifically, we refined the explanation of the experimental setup, data processing workflow, and interpretation of the CV and DPV results, emphasizing that the conclusions are directly derived from the analyzed data. These revisions improve clarity and technical transparency without altering the underlying results or conclusions of the study.

3. Has the statistical analysis been performed appropriately and rigorously?

Reviewer #2: Yes

Reviewer #3: No

Response

We thank the reviewers for their evaluation. In this study, the data analysis is based on established electrochemical signal processing methods rather than inferential statistical testing. We have clarified this distinction in the revised manuscript by explicitly describing the analytical procedures used for baseline correction, peak detection, and signal interpretation for CV and DPV data. These methods were applied consistently across all measurements and provide an appropriate and rigorous basis for the conclusions drawn

4. Have the authors made all data underlying the findings in their manuscript fully available?

Reviewer #2: Yes

Reviewer #3: Yes

Response

We thank the reviewers for their confirmation. All data underlying the findings described in the manuscript, including raw voltammetry measurement files and processed analysis outputs, are fully available through the repository indicated in the Data Availability Statement.

5. Is the manuscript presented in an intelligible fashion and written in standard English?

Reviewer #2: Yes

Reviewer #3: No

Response

We thank the reviewers for their feedback. In response to Reviewer #3’s comment, we have carefully revised and polished the manuscript to improve clarity, consistency, and readability in standard scientific English. Typographical and grammatical errors have been corrected, and several sentences have been rephrased to enhance precision and flow, without changing the scientific content of the manuscript.

6. Review Comments to the Author

Reviewer #2: I read the revised manuscript and found that it addresses previous comments well and incorporates the revisions. I, however, am not sure whether the style of citing the reference in the text is ..... [1]. or ...... . [1]. Both styles are used in the manuscript; it should be corrected.

Reviewer #3: English of manuscript is below the required standards and needs a careful polish.

Section Potentiostat and voltammetry needs a careful revision. Why SPCE electrode is used? Ag/AgCl paste!!!

what is the reference electrode? non-useful parts eg. definitions needs to be removed.

the title of Anshori paper in page 5 must be deleted. what is the baseline correction system? Conclusion section must be re-prepared as more informative. Statistical analysis section or methods?

Response

We thank the reviewers for their constructive comments.

In response to Reviewer #2, we have standardized the in-text citation format throughout the manuscript to ensure consistency with the journal’s reference style.

In response to Reviewer #3, we have carefully polished the English language to meet the journal’s standards. The Potentiostat and Voltammetry section has been revised to improve clarity, explicitly justify the use of the SPCE, and clearly describe the reference electrode configuration. Non-essential definitions have been removed, and the title of the cited work by Anshori has been deleted from the main text. We have also clarified the baseline correction workflow, revised the Conclusion section to be more informative, and explicitly positioned the analytical procedures within the Methods section to address concerns regarding statistical analysis. These revisions improve clarity and presentation without altering the scientific content of the manuscript.

7. PLOS authors have the option to publish the peer review history of their article (what does this mean?). If published, this will include your full peer review and any attached files.

Do you want your identity to be public for this peer review? For information about this choice, including consent withdrawal, please see our Privacy Policy.

Reviewer #2: Yes: Jagadeesh Bhattarai

Reviewer #3: No

Response

We acknowledge the reviewers’ choices regarding the publication of peer review history and comply with the journal’s policy on peer review transparency and anonymity.

---

## [Editor Report · Decision Letter 3]

15 Apr 2026

Algorithm-Assisted Interpretation of Cyclic and Differential Pulse Voltammetry for Cardiac Troponin Detection

PONE-D-24-56032R3

Dear Dr. Sunindyo,

We’re pleased to inform you that your manuscript has been judged scientifically suitable for publication and will be formally accepted for publication once it meets all outstanding technical requirements.

Kind regards,

Abbas Farmany

Academic Editor

PLOS One
---

## [Editor Report · Acceptance letter]

PONE-D-24-56032R3

PLOS One

Dear Dr. Sunindyo,

I'm pleased to inform you that your manuscript has been deemed suitable for publication in PLOS One. Congratulations! Your manuscript is now being handed over to our production team.

Kind regards,

on behalf of

Dr. Abbas Farmany

Academic Editor

PLOS One